# Replay-Guided Adversarial Environment Design

**Minqi Jiang**[*]
UCL, FAIR

**Michael Dennis**[*]
UC Berkeley

**Jack Parker-Holder**
University of Oxford

**Jakob Foerster**
FAIR

**Edward Grefenstette**
UCL, FAIR

**Tim Rocktäschel**
UCL, FAIR

## Abstract

Deep reinforcement learning (RL) agents may successfully generalize to new settings if trained on an appropriately diverse set of environment and task configurations. Unsupervised Environment Design (UED) is a promising self-supervised RL paradigm, wherein the free parameters of an underspecified environment are automatically adapted during training to the agent's capabilities, leading to the emergence of diverse training environments. Here, we cast Prioritized Level Replay (PLR), an empirically successful but theoretically unmotivated method that selectively samples randomly-generated training levels, as UED. We argue that by curating completely random levels, PLR, too, can generate novel and complex levels for effective training. This insight reveals a natural class of UED methods we call Dual Curriculum Design (DCD). Crucially, DCD includes both PLR and a popular UED algorithm, PAIRED, as special cases and inherits similar theoretical guarantees. This connection allows us to develop novel theory for PLR, providing a version with a robustness guarantee at Nash equilibria. Furthermore, our theory suggests a highly counterintuitive improvement to PLR: by stopping the agent from updating its policy on uncurated levels (training on *less* data), we can improve the convergence to Nash equilibria. Indeed, our experiments confirm that our new method, $PLR^{\perp}$, obtains better results on a suite of out-of-distribution, zero-shot transfer tasks, in addition to demonstrating that $PLR^{\perp}$ improves the performance of PAIRED, from which it inherited its theoretical framework.

## 1 Introduction

While deep reinforcement learning (RL) approaches have led to many successful applications in challenging domains like Atari [21], Go [35], Chess [36], Dota [4], and StarCraft [40] in recent years, deep RL agents still prove to be brittle, often failing to transfer to environments only slightly different from those encountered during training [44, 9]. To ensure learning of robust and well-generalizing policies, agents must train on sufficiently diverse and informative variations of environments (e.g. see Section 3.1 of [8]). However, it is not always feasible to specify an appropriate training distribution or a generator thereof. Agents may therefore benefit greatly from methods that automatically adapt the distribution over environment variations throughout training [10, 17]. Throughout this paper we will call a particular environment instance or configuration (e.g. an arrangement of blocks, race tracks, or generally any of the environment's constituent entities) a *level*.

Two recent works [10, 17] have sought to empirically demonstrate this need for a more targeted agent-adaptive mechanism for selecting levels on which to train RL agents, so to ensure efficient learning and generalization to unseen levels—as well as to provide methods implementing such mechanisms. The first method, Protagonist Antagonist Induced Regret Environment Design (PAIRED) [10],

---

[*]Equal contribution. Correspondence to `msj@fb.com` and `michael_dennis@berkeley.edu`.

35th Conference on Neural Information Processing Systems (NeurIPS 2021).

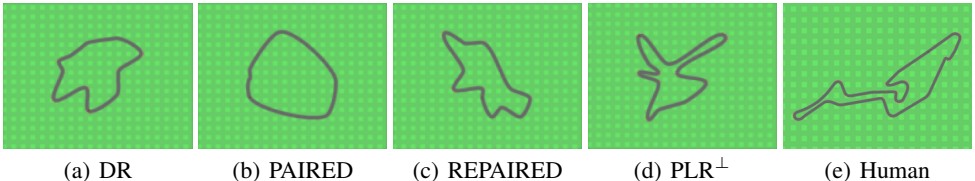

| (a) DR | (b) PAIRED | (c) REPAIRED | (d) PLR$^\perp$ | (e) Human |

Figure 1: Randomly drawn samples of CarRacing tracks produced by different methods. (a) Domain Randomization (DR) produces tracks of average complexity, with few sharp turns. (b) PAIRED often overexploits the difference in the students, leading to simple tracks that incidentally favor the antagonist. (c) REPAIRED mitigates this degeneracy, recovering track complexity. (d) PLR$^\perp$ selects the most challenging randomly generated tracks, resulting in tracks that more closely resemble human-designed tracks, such as (e) the Nürburgring Grand Prix.

introduces a self-supervised RL paradigm called Unsupervised Environment Design (UED). Here, an environment generator (a *teacher*) is co-evolved with a *student* policy that trains on levels actively proposed by the teacher, leading to a form of adaptive curriculum learning. The aim of this coevolution is for the teacher to gradually learn to generate environments that exemplify properties of those that might be encountered at deployment time, and for the student to simultaneously learn a good policy that enables zero-shot transfer to such environments. PAIRED's specific adversarial approach to environment design ensures a useful robustness characterization of the final student policy in the form of a minimax regret guarantee [31]—assuming that its underlying teacher-student multi-agent system arrives at a Nash equilibrium [NE, 24]. In contrast, the second method, Prioritized Level Replay (PLR) [17], embodies an alternative form of dynamic curriculum learning that does not assume control of level generation, but instead, the ability to selectively replay existing levels. PLR tracks levels previously proposed by a black-box environment generator, and for each, estimates the agent's learning potential in that level, in terms of how useful it would be to gather new experience from that level again in the future. The PLR algorithm exploits these scores to adapt a schedule for revisiting or *replaying* levels to maximize learning potential. PLR has been shown to produce scalable and robust results, improving both sample complexity of agent training and the generalization of the learned policy in diverse environments. However, unlike PAIRED, PLR is motivated with heuristic arguments and lacks a useful theoretical characterization of its learning behavior.

In this paper, we argue that PLR is, in and of itself, an effective form of UED: Through curating even randomly generated levels, PLR can generate novel and complex levels for learning robust policies. This insight leads to a natural class of UED methods which we call *Dual Curriculum Design* (DCD). In DCD, a student policy is challenged by a team of two co-evolving teachers. One teacher actively generates new, challenging levels, while the other passively curates existing levels for replaying, by prioritizing those estimated to be most suitably challenging for the student. We show that PAIRED and PLR are distinct members of the DCD class of algorithms and prove in Section 3 that all DCD algorithms enjoy similar minimax regret guarantees to that of PAIRED.

We make use of this result to provide the first theoretical characterization of PLR, which immediately suggests a simple yet highly counterintuitive adjustment to PLR: By only training on trajectories in replay levels, PLR becomes provably robust at NE. We call this resulting variant PLR$^\perp$ (Section 4). From this perspective, PLR effectively performs level design in a diametrically opposite manner to PAIRED—through prioritized selection rather than active generation. A second corollary to the provable robustness of DCD algorithms shows that PLR$^\perp$ can be extended to make use of the PAIRED teacher as a level generator while preserving the robustness guarantee of PAIRED, resulting in a method we call *Replay-Enhanced PAIRED* (REPAIRED) (Section 5). We hypothesize that in this arrangement, PLR$^\perp$ plays a complementary role to PAIRED in robustifying student policies.

Our experiments in Section 6 investigate the learning dynamics of PLR$^\perp$, REPAIRED, and their replay-free counterparts on a challenging maze domain and a novel continuous control UED setting based on the popular CarRacing environment [5]. In both of these highly distinct settings, our methods provide significant improvements over PLR and PAIRED, producing agents that can perform out-of-distribution (OOD) generalization to a variety of human designed mazes and Formula 1 tracks.

In summary, we present the following contributions: (i) We establish a common framework, Dual Curriculum Design, that encompasses PLR and PAIRED. This allows us to develop new theory, which provides the first robustness guarantees for PLR at NE as well as for REPAIRED, which

augments PAIRED with a PLR-based replay mechanism. (ii) Crucially, our theory suggests a highly counterintuitive improvement to PLR: the convergence to NE should be assisted by training on less data when using PLR—namely by only taking gradient updates from data that originates from the PLR buffer, using the samples from the environment distribution only for computing the prioritization of levels in the buffer. (iii) Our experiments in a maze domain and a novel car racing domain show that our methods significantly outperform their replay-free counterparts in zero-shot generalization. We open source our methods at https://github.com/facebookresearch/dcd.

## 2 Background

### 2.1 Unsupervised Environment Design

Unsupervised Environment Design (UED), as introduced by [10], is the problem of automatically designing a distribution of environments that adapts to the learning agent. UED is defined in terms of an *Underspecified POMDP* (UPOMDP), given by $\mathcal{M} = \langle A, O, \Theta, S^{\mathcal{M}}, \mathcal{T}^{\mathcal{M}}, \mathcal{I}^{\mathcal{M}}, \mathcal{R}^{\mathcal{M}}, \gamma \rangle$, where $A$ is a set of actions, $O$ is a set of observations, $S$ is a set of states, $\mathcal{T} : S \times A \times \Theta \to \mathbf{\Delta}(S)$ is a transition function, $\mathcal{I} : S \to O$ is an observation (or inspection) function, $\mathcal{R} : S \to \mathbb{R}$ is a reward function, and $\gamma$ is a discount factor. This definition is identical to a POMDP with the addition of $\Theta$ to represent the free-parameters of the environment. These parameters can be distinct at every time step and incorporated into the transition function $\mathcal{T}^{\mathcal{M}} : S \times A \times \Theta \to \mathbf{\Delta}(S)$. For example, $\Theta$ could represent the possible positions of obstacles in a maze. We will refer to the environment resulting from a fixed $\theta \in \Theta$ as $\mathcal{M}_\theta$, or with a slight abuse of notation, simply $\theta$ when clear from context. We define the value of $\pi$ in $\mathcal{M}_\theta$ to be $V^\theta(\pi) = \mathbb{E}[\sum_{i=0}^{T} r_t \gamma^t]$ where $r_t$ are the rewards attained by $\pi$ in $\mathcal{M}_\theta$. Aligning with terminology from [17], we refer to a fully-specified environment as a *level*.

### 2.2 Protagonist Antagonist Induced Regret Environment Design

Protagonist Antagonist Induced Regret Environment Design [PAIRED, 10] presents a UED approach consisting of simultaneously training agents in a three player game: the protagonist $\pi_A$ and the antagonist $\pi_B$ are trained in environments generated by the teacher $\tilde{\theta}$. The objective of this game is defined by $U(\pi_A, \pi_B, \tilde{\theta}) = \mathbb{E}_{\theta \sim \tilde{\theta}}[\text{REGRET}^\theta(\pi_A, \pi_B)]$, where regret is defined by $\text{REGRET}^\theta(\pi_A, \pi_B) = V^\theta(\pi_B) - V^\theta(\pi_A)$. The protagonist and antagonist are both trained to maximize their discounted environment returns while the teacher is trained to maximize $U$. Note that by maximizing regret, the teacher is disincentivized from generating unsolvable levels, which will have a maximum regret of $0$. As shorthand, we will sometimes refer to the protagonist and antagonist jointly as the *student agents*. The counterclockwise loop beginning at the student agents in Figure 2 summarizes this approach, with the students being both the protagonist and antagonist.

As both student agents grow more adept at solving different levels, the teacher continues to adapt its level designs to exploit the weaknesses of the protagonist in relation to the antagonist. As this dynamic unfolds, PAIRED produces an emergent curriculum of progressively more complex levels along the boundary of the protagonist's capabilities. PAIRED is a creative method in the sense that the teacher may potentially generate an endless sequence of novel levels. However, as the teacher only adapts through gradient updates, it is inherently slow to adapt to changes in the student policies.

### 2.3 Prioritized Level Replay

Prioritized Level Replay [PLR, 17] is an active-learning strategy shown to improve a policy's sample efficiency and generalization to unseen levels when training and evaluating on levels from a common UPOMDP, typically implemented as a seeded simulator. PLR maintains a level buffer $\mathbf{\Lambda}$ of the top $K$ visited levels with highest learning potential as estimated by the time-averaged L1 value loss of the learning agent over the last episode on each level. At the start of each training episode, with some predefined replay probability $p$, PLR uses a bandit to sample the level from $\mathbf{\Lambda}$ to maximize the estimated learning potential; otherwise, with probability $1 - p$, PLR samples a new level from the simulator. In contrast to the generative but slow-adapting PAIRED, PLR does not create new levels, but instead, acts as a fast-adapting curation mechanism for selecting the next training level among previously encountered levels. Also unlike PAIRED, PLR does not provide a robustness guarantee. By extending the theoretical foundation of PAIRED to PLR, we will show how PLR can be modified

to provide a robustness guarantee at NE, as well as how PAIRED can exploit PLR's complementary curation to quickly switch among generated levels to maximize the student's regret.

# 3   The Robustness of Dual Curriculum Design

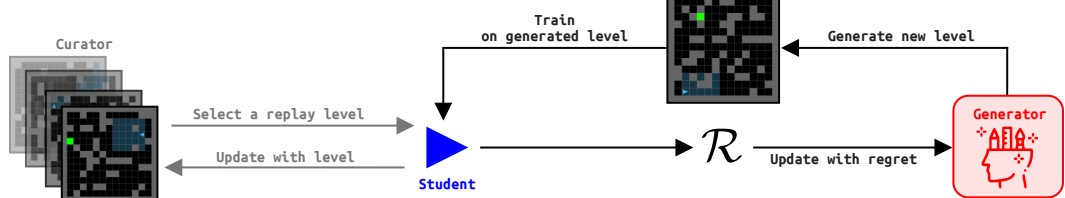

Figure 2: Overview of Dual Curriculum Design (DCD). The student learns in the presence of two co-adapting teachers that aim to maximize the student's regret: The generator teacher designs new levels to challenge the agent, and the curator teacher prioritizes a set of levels already created, selectively sampling them for replay.

The previous approaches of PAIRED and PLR reveal a natural duality: Approaches that gradually learn to generate levels like PAIRED, and methods which cannot generate levels, but instead, quickly curate existing ones, like PLR. This duality suggests combining slow level generators with fast level curators. We call this novel class of UED algorithms Dual Curriculum Design (DCD). For instance, PLR can be seen as curator with a prioritized sampling mechanism with a random generator, while PAIRED, as a regret-maximizing generator without a curator. DCD can further consider Domain Randomization (DR) as a degenerate case of a random level generator without a curator.

To theoretically analyze this space of methods, we model DCD as a three player game among a student agent and two teachers called the *dual curriculum game*. However, to formalize this game, we must first formalize the single-teacher setting: Suppose the UPOMDP is clear from context. Then, given a utility function for a single teacher, $U_t(\pi, \theta)$, we can naturally define the *base game* between the student $s$ and teacher $t$ as $G = \langle S = S_s \times S_t, U = U_s \times U_t \rangle$, where $S_s = \Pi$ is the strategy set of the student, $S_t = \Theta$ is the strategy set of the teacher, and $U_s(\pi, \theta) = V^\theta(\pi)$ is the utility function of the student. In Sections 4 and 5, we will study settings corresponding to different choices of utility functions for the teacher agents, namely the maximum-regret objective $U_t^R(\pi, \theta)$ and the uniform objective $U_t^U(\pi, \theta)$. These two objectives are defined as follows (for any constant $C$):

$$U_t^R(\pi, \theta) = \underset{\pi^* \in \Pi}{\operatorname{argmax}} \{V^\theta(\pi^*) - V^\theta(\pi)\} \tag{1}$$

$$U_t^U(\pi, \theta) = C \tag{2}$$

In the dual curriculum game $\overline{G}$, the first teacher plays the game with probability $p$, and the second, with probability $(1-p)$—or more formally, $\overline{G} = \langle \overline{S} = S_s \times S_t \times S_t, U = \overline{U}_s \times \overline{U}_t^1 \times \overline{U}_t^2 \rangle$, where the utility functions for the student and two teachers respectively, $\overline{U}_s, \overline{U}_t^1, \overline{U}_t^2$, are defined as follows:

$$\overline{U}_t^1(\pi, \theta^1, \theta^2) = pU_t^1(\pi, \theta^1) \tag{3}$$

$$\overline{U}_t^2(\pi, \theta^1, \theta^2) = (1-p)U_t^2(\pi, \theta^2) \tag{4}$$

$$\overline{U}_s(\pi, \theta^1, \theta^2) = pU_s(\pi, \theta^1) + (1-p)U_s(\pi, \theta^2) \tag{5}$$

Our main theorem is that NE in the dual curriculum game are approximate NE of both the base game for either of the original teachers and the base game with a teacher maximizing the joint-reward of $pU_t^1 + (1-p)U_t^2$, where the quality of the approximations depends on the mixing probability $p$.

**Theorem 1.** *Let $B$ be the maximum difference between $U_t^1$ and $U_t^2$, and let $(\pi, \theta^1, \theta^2)$ be a NE for $\overline{G}$. Then $(\pi, p\theta^1 + (1-p)\theta^2)$ is an approximate NE for the base game with either teacher or for a teacher optimizing their joint objective. More precisely, it is a $2Bp(1-p)$-approximate NE when $U_t = pU_t^1 + (1-p)U_t^2$, a $2B(1-p)$-approximate NE when $U_t = U_t^1$, and a $2Bp$-approximate NE when $U_t = U_t^2$.*

The intuition behind this theorem is that, since the two teachers do not affect each other's behavior, their best response to a fixed $\pi_s$ is to choose a strategy $\theta$ that maximizes $U_t^1$ and $U_t^2$ respectively.

Moreover, the two teachers' strategies can be viewed as a single combined strategy for the base game with the joint-objective, or with each teacher's own objective. In fact, the teachers provide an approximate best-response to each case of the base game simply by playing their individual best responses. Thus, when we reach a NE of the dual curriculum game, the teachers arrive at approximate best responses for both the base game with the joint objective and with their own objectives, meaning they are also in an approximate NE of the base game with either teacher. The full details of this proof are outlined in Appendix A.

## 4 Robustifying PLR

In this section, we provide theoretical justification for the empirically observed effectiveness of PLR, and in the process, motivate a counterintuitive adjustment to the algorithm.

---

**Algorithm 1:** Robust PLR (PLR$^{\perp}$)

---

Randomly initialize policy $\pi(\phi)$ and an empty level buffer, $\mathbf{\Lambda}$ of size $K$.
**while** *not converged* **do**
    Sample replay-decision Bernoulli, $d \sim P_D(d)$
    **if** $d = 0$ **then**
        Sample level $\theta$ from level generator
        Collect $\pi$'s trajectory $\tau$ on $\theta$, with a stop-gradient $\phi_{\perp}$         *i.e. Suppress policy update*
    **else**
        Use PLR to sample a replay level from the level store, $\theta \sim \mathbf{\Lambda}$
        Collect policy trajectory $\tau$ on $\theta$ and update $\pi$ with rewards $\boldsymbol{R}(\tau)$
    **end**
    Compute PLR score, $S = \mathbf{score}(\tau, \pi)$
    Update $\mathbf{\Lambda}$ with $\theta$ using score $S$
**end**

---

### 4.1 Achieving Robustness Guarantees with PLR

PLR provides strong empirical gains in generalization, but lacks any theoretical guarantees of robustness. One step towards achieving such a guarantee is to replace its L1 value-loss prioritizaton with a regret prioritization, using the methods we discuss in Section 4.2: While L1 value loss may be good for quickly training the value function, it can bias the long-term training behavior toward high-variance policies. However, even with this change, PLR holds weaker theoretical guarantees because the random generating teacher can bias the student away from minimax regret policies and instead, toward policies that sacrifice robustness in order to excel in unstructured levels. We formalize this intuitive argument in the following corollary of Theorem 1.

**Corollary 1.** *Let $\overline{G}$ be the dual curriculum game in which the first teacher maximizes regret, so $U_t^1 = U_t^R$, and the second teacher plays randomly, so $U_t^2 = U_t^U$. Let $V^{\theta}(\pi)$ be bounded in $[B^-, B^+]$ for all $\theta, \pi$. Further, suppose that $(\pi, \theta^1, \theta^2)$ is a Nash equilibrium of $\overline{G}$. Let $R^* = \min_{\pi_A \in \Pi}\{\max_{\theta, \pi_B \in \Theta, \Pi}\{\text{REGRET}^{\theta}(\pi_A, \pi_B)\}\}$ be the optimal worst-case regret. Then $\pi$ is $2(B^+ - B^-)(1 - p)$ close to having optimal worst-case regret, or formally, $\max_{\theta, \pi_B \in \Theta, \Pi}\{\text{REGRET}^{\theta}(\pi_A, \pi)\} \geq R^* - 2(B^+ - B^-)(1 - p)$. Moreover, there exists environments for all values of $p$ within a constant factor of achieving this bound.*

The proof of Corollary 1 follows from a direct application of Theorem 1 to show that a NE of $\overline{G}$ is an approximate NE for the base game of the first teacher, and through constructing a simple example where the student's best response in $\overline{G}$ fails to attain the minimax regret in $G$. These arguments are described in full in Appendix A. This corollary provides some justification for why PLR improves robustness of the equilibrium policy, as it biases the resulting policy toward a minimax regret policy. However, it also points a way towards further improving PLR: If the probability $p$ of using a teacher-generated level directly was set to 0, then in equilibrium, the resulting policy converges to a minimax regret policy. Consequently, we arrive at the counterintuitive idea of avoiding gradient updates from trajectories collected from randomly sampled levels, to ensure that at NE, we find a minimax regret policy. From a robustness standpoint, it is therefore optimal to train on less data. The modified PLR algorithm PLR$^{\perp}$ with this counterintuitive adjustment is summarized in Algorithm 1, in which this small change relative to the original algorithm is highlighted in blue.

### 4.2 Estimating Regret

In general, levels may differ in maximum achievable returns, making it impossible to know the true regret of a level without access to an oracle. As the L1 value loss typically employed by PLR does not generally correspond to regret, we turn to alternative scoring functions that better approximate regret. Two approaches, both effective in practice, are discussed below.

**Positive Value Loss** Averaging over all transitions with positive value loss amounts to estimating regret as the difference between maximum achieved return and predicted return on an episodic basis. However, this estimate is highly biased, as the value targets are tied to the agent's current, potentially suboptimal policy. As it only considers positive value losses, this scoring function leads to optimistic sampling of levels with respect to the current policy. When using GAE [33] to estimate bootstrapped value targets, this loss takes the following form, where $\lambda$ and $\gamma$ are the GAE and MDP discount factors respectively, and $\delta_t$, the TD-error at timestep $t$:

$$\frac{1}{T}\sum_{t=0}^{T}\max\left(\sum_{k=t}^{T}(\gamma\lambda)^{k-t}\delta_k,0\right).$$

**Maximum Monte Carlo (MaxMC)** We can mitigate some of the bias of the positive value loss by replacing the value target with the highest return achieved on the given level so far during training. By using this maximal return, the regret estimates no longer depend on the agent's current policy. This estimator takes the simple form of $(1/T)\sum_{t=0}^{T}R_{\max}-V(s_t)$. In our dense-reward experiments, we compute this score as the difference between the maximum achieved return and $V(s_0)$.

## 5  Replay-Enhanced PAIRED (REPAIRED)

We can replace the random generator teacher used by $\text{PLR}^{\perp}$ with the PAIRED teacher. This extension entails a second student agent, the antagonist, also equipped with its own PLR level buffer. In each episode, with probability $p$, the students evaluate their performances (but do not train) on a newly generated level and, with probability $1 - p$, train on a level sampled from each student's own regret-prioritizing PLR buffer. Training only on the highest regret levels should mitigate inefficiencies in the PAIRED teacher's optimization procedure. We refer to this extension as *Replay-Enhanced PAIRED* (REPAIRED). An overview of REPAIRED is provided by black arrows in Figure 2, with the students being the protagonist and antagonist, while the full pseudocode is outlined in Appendix B.

Since $\text{PLR}^{\perp}$ and PAIRED both promote regret in equilibrium, it would be reasonable to believe that the combination of the two does the same. A straightforward corollary of Theorem 1, which we describe in Appendix 1, shows that, in a theoretically ideal setting, combining these two algorithms as is done in REPAIRED indeed finds minimax regret strategies in equilibrium.

**Corollary 2.** *Let $\overline{G}$ be the dual curriculum game in which both teachers maximize regret, so $U_t^1 = U_t^2 = U_t^R$. Further, suppose that $(\pi, \theta^1, \theta^2)$ is a Nash equilibrium of $\overline{G}$. Then, $\pi \in \text{argmin}_{\pi_A \in \Pi}\{\max_{\theta,\pi_B \in \Theta,\Pi}\{\text{REGRET}^{\theta}(\pi_A, \pi_B)\}\}$.*

This result gives us some amount of assurance that, if our method arrives at NE, then the protagonist has converged to a minimax regret strategy, which has the benefits outlined in [10]: Since a minimax regret policy solves all solvable environments, whenever this is possible and sufficiently well-defined, we should expect policies resulting from the equilibrium behavior of REPAIRED to be robust and versatile across all environments in the domain.

## 6  Experiments

Our experiments firstly aim to (1) assess the empirical performance of the theoretically motivated $\text{PLR}^{\perp}$, and secondly, seek to better understand the effect of replay on unsupervised environment design, specifically (2) its impact on the zero-shot generalization performance of the induced student policies, and (3) the complexity of the levels designed by the teacher. To do so, we compare PLR and REPAIRED against their replay-free counterparts, DR and PAIRED, in the two highly distinct settings of discrete control with sparse rewards and continuous control with dense rewards. We provide environment descriptions alongside model and hyperparameter choices in Appendix D.

## 6.1 Partially-Observable Navigation

Each navigation level is a partially-observable maze requiring student agents to take discrete actions to reach a goal and receive a sparse reward. Our agents use PPO [34] with an LSTM-based recurrent policy to handle partial observability. Before each episode, the teacher designs the level in this order: beginning with an empty maze, it places one obstructing block per time step up to a predefined block budget, and finally places the agent followed by the goal.

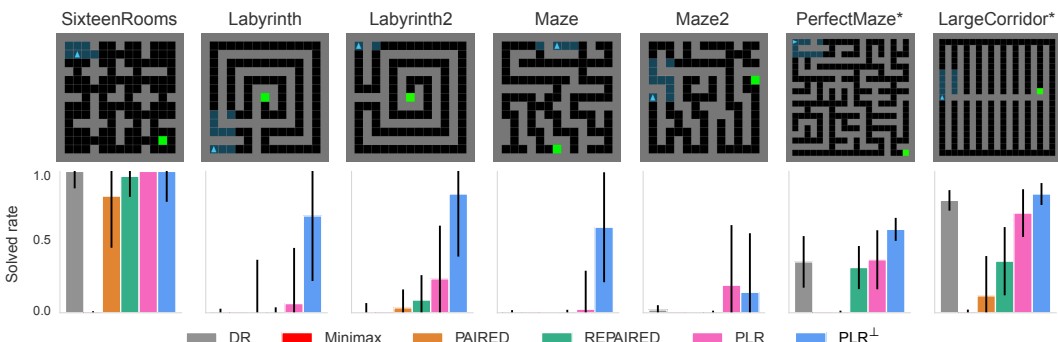

Figure 3: Zero-shot transfer performance in challenging test environments after 250M training steps. The plots show median and interquartile range of solved rates over 10 runs. An asterisk (*) next to the maze name indicates the maze is procedurally-generated, and thus each attempt corresponds to a random configuration of the maze.

**Zero-Shot Generalization** We train policies with each method for 250M steps and evaluate zero-shot generalization on several challenging OOD environments, in addition to levels from the full distribution of two procedurally-generated environments, PerfectMaze and LargeCorridor. We also compare against DR and minimax baselines. Our results in Figure 3 and 4 show that PLR$^\perp$ and REPAIRED both achieve greater sample-efficiency and zero-shot generalization than their replay-free counterparts. The improved test performance achieved by PLR$^\perp$ over both DR and PLR when trained for an equivalent number of gradient updates, aggregated over all test mazes, is statistically significant ($p < 0.05$), as is the improved test performance of REPAIRED over PAIRED. Well before 250 million steps, both PLR and PLR$^\perp$ significantly outperform PAIRED after 3 billion training steps, as reported in [10]. Further, both PLR variants lead to policies exhibiting greater zero-shot transfer than the PAIRED variants. Notably, the PLR$^\perp$ agent learns to solve mazes via an approximate right-hand rule. Table 2 in Appendix C.1 reports performance across all test mazes. The success of designing regret-maximizing levels via random search (curation) over learning a generator with RL suggests that for some UPOMDPs, the regret landscape, as a function of the free parameters $\theta$, has a low effective dimensionality [3]. Foregoing gradient-based learning in favor of random search may then lead to faster adaptation to the changing regret landscape, as the policy evolves throughout training.

**Emergent Complexity** As the student agents improve, the teachers must generate more challenging levels to maintain regret. We measure the resultant emergent complexity by tracking the number of blocks in each level and the shortest path length to the goal (where unsolvable levels are assigned a length of 0). These results, summarized in Figure 4, show that PAIRED slowly adapts the complexity over training while REPAIRED initially quickly grows complexity, before being overtaken by PAIRED. This more rapid onset of complexity may be

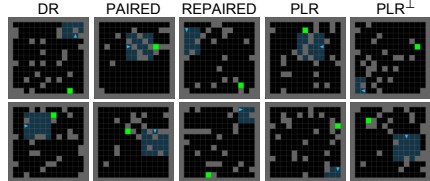

Figure 5: Examples of emergent structures generated by each method.

due to REPAIRED's fast replay mechanism, and the long-term slowdown relative to PAIRED may be explained by its less frequent gradient updates. Our results over an extended training period in Appendix C confirm that both PAIRED and REPAIRED slowly increase complexity over time, eventually matching that attained in just a fraction of the number of gradient steps by PLR and PLR$^\perp$. This result shows that random search is surprisingly efficient at continually discovering levels of increasing complexity, given an appropriate curation mechanism such as PLR. Figure 5 shows that, similar to methods with a regret-maximizing teacher, PLR finds levels exhibiting complex structure.

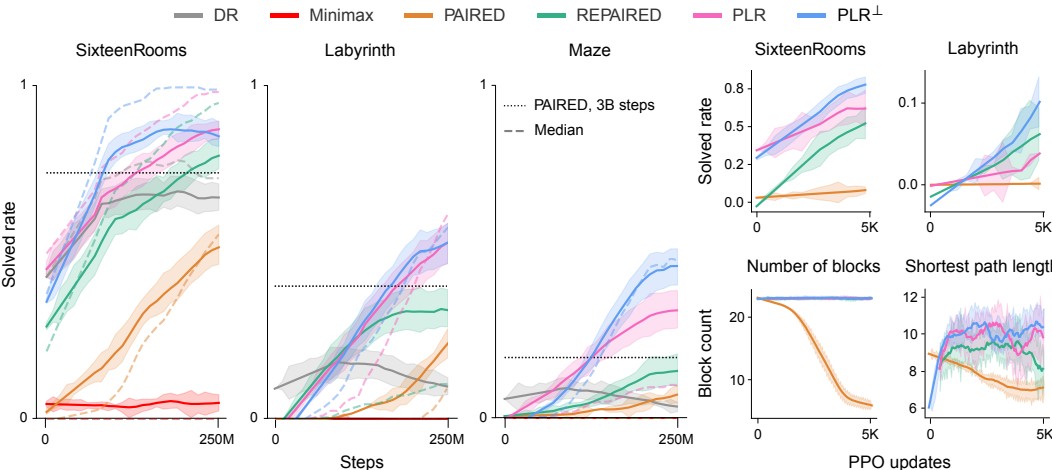

Figure 4: Zero-shot transfer performance during training for PAIRED and REPAIRED variants. The plots show mean and standard error across 10 runs. The dotted lines mark the mean performance of PAIRED after 3B training steps, as reported in [10], while dashed lines indicate median returns.

## 6.2 Pixel-Based Car Racing with Continuous Control

To test the versatility and scalability of our methods, we turn to an extended version of the CarRacing environment from OpenAI Gym [5]. This environment entails continuous control with dense rewards, a 3-dimensional action space, and partial, pixel observations, with the goal of driving a full lap around a track. To enable UED of any closed-loop track, we reparameterize CarRacing to generate tracks as Bézier curves [22] with arbitrary control points. The teacher generates levels by choosing a sequence of up to 12 control points, which uniquely defines a Bézier track within specific, predefined curvature constraints. After 5M steps of training, we test the zero-shot transfer performance of policies trained by each method on 20 levels replicating official human-designed Formula One (F1) tracks (see Figure 19 in the Appendix for a visualization of the tracks). Note that these tracks are significantly OOD, as they cannot be defined with just 12 control points. In Figure 6 we show the progression of zero-shot transfer performance for the original CarRacing environment, as well as three F1 tracks of varying difficulty, while also including the final performance on the full F1 benchmark. For the final performance, we also evaluated the state-of-the-art CarRacing agent from [38] on our new F1 benchmark.

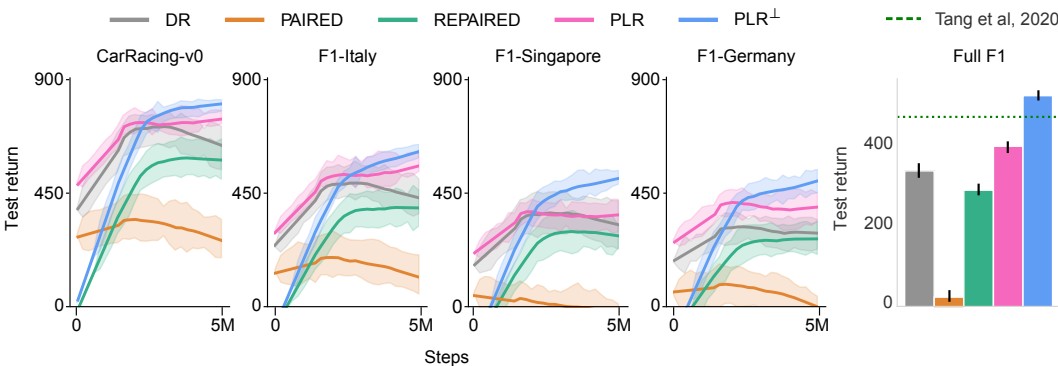

Figure 6: Zero-shot transfer performance. Plots show mean and standard error over 10 runs.

Unlike in the sparse, discrete navigation setting, we find DR leads to moderately successful policies for zero-shot transfer in CarRacing. Dense rewards simplify the learning problem and random Bezier tracks occasionally contain the challenges seen in F1 tracks, such as hairpin turns and observations showing parallel tracks due to high local curvature. Still, we see that policies trained by selectively sampling tracks to maximize regret significantly outperform those trained by uniformly sampling

from randomly generated tracks, in terms of zero-shot transfer to the OOD F1 tracks. Remarkably, with a replay rate of 0.5, $PLR^{\perp}$ sees statistically significant ($p < 0.001$) gains over PLR in zero-shot performance over the full F1 benchmark, despite directly training on only half the rollout data using half as many gradient updates. Once again, we see that random search with curation via PLR produces a rich selection of levels and an effective curriculum.

We also observe that PAIRED struggles to train a robust protagonist in CarRacing. Specifically, PAIRED overexploits the relative strengths of the antagonist over the protagonist, finding curricula that steer the protagonist towards policies that ultimately perform poorly even on simple tracks, leading to a gradual reduction in level complexity. We present training curves revealing this dynamic in Appendix C. As shown in Figure 6, REPAIRED mitigates this degeneracy substantially, though not completely, inducing a policy that significantly outperforms PAIRED ($p < 0.001$) in mean performance on the full F1 benchmark, but underperforms DR. Notably, $PLR^{\perp}$ exceeds the performance of the state-of-the-art AttentionAgent [38], despite not using a self-attention policy and training on less than 0.25% of the number of environment steps in comparison. These gains come purely from the induced curriculum. Figure 17 in Appendix C further reveals that $PLR^{\perp}$ produces CarRacing policies that tend to achieve higher minimum returns on average compared to the baseline methods, providing further evidence of the benefits of the minimax regret property coupled with a fast replay-based mechanism for efficiently finding high-regret levels.

## 7 Related Work

In inducing parallel curricula, DCD follows a rich lineage of curriculum learning methods [2, 32, 28, 23]. Many previous curriculum learning algorithms resemble the curator in DCD, sharing similar underlying selective-sampling mechanisms as $PLR^{\perp}$. Most similar is TSCL [19], which prioritizes levels based on return rather than value loss, and has been shown to overfit to training levels in some settings [17]. In our setting, replayed levels can be viewed as past strategies from a level-generating teacher. This links our replay-based methods to fictitious self-play [FSP, 13], and more closely, Prioritized FSP [40], which selectively samples opponents based on historic win ratios.

Recent approaches that make use of a generating adversary include Asymmetric Self-Play [37, 26], wherein one agent proposes tasks for another in the form of environment trajectories, and AMIGo [6], wherein the teacher is rewarded for proposing reachable goals. While our methods do not presuppose a goal-based setting, others have made progress here using generative modeling [12, 29], latent skill learning [14], and exploiting model disagreement [45]. These methods are less generally applicable than $PLR^{\perp}$, and unlike our DCD methods, they do not provide well-principled robustness guarantees.

Other recent algorithms can be understood as forms of UED and like DCD, framed in the lens of decision theory. POET [41, 42], a coevolutionary approach [27], uses a population of *minimax* (rather than minimax regret) adversaries to construct terrain for a BipedalWalker agent. In contrast to our methods, POET requires training a large population of both agents and environments and consequently, a sizable compute overhead. APT-Gen [11] also procedurally generates tasks, but requires access to target tasks, whereas our methods seek to improve zero-shot transfer.

The DCD framework also encompasses adaptive domain randomization methods [DR, 20, 15], which have seen success in assisting sim2real transfer for robotics [39, 16, 1, 25]. DR itself is subsumed by procedural content generation [30], for which UED and DCD may be seen as providing a formal, decision-theoretic framework, enabling development of provably optimal algorithms.

## 8 Discussion

We established a novel connection between PLR and minimax regret UED approaches like PAIRED, by developing the theory of Dual Curriculum Design (DCD). In this setting, a student policy is challenged by a team of two co-adapting, regret-maximizing teachers: one, a generator that creates new levels, and the other, a curator that selectively samples previously generated levels for replay. This view unifies PLR and PAIRED, which are both instances of DCD. Our theoretical results on DCD then enabled us to prove that PLR attains a minimax regret policy at NE, thereby providing the first theoretical characterization of the robustness of PLR. Notably our theory leads to the counterintuitive result that PLR can be made provably robust by training on less data, specifically, by only using

the trajectories on levels sampled for replay. In addition, we developed Replay-Enhanced PAIRED (REPAIRED), which extends the selective replay-based updates of $PLR^{\perp}$ to PAIRED, and proved it shares the same robustness guarantee at NE. Empirically, in two highly distinct environments, we found that $PLR^{\perp}$ significantly improves zero-shot generalization over PLR, and REPAIRED, over PAIRED. As our methods solely modify the order of levels visited during training, they can, in principle, be combined with many other RL methods to yield potentially orthogonal improvements in sample-efficiency and generalization.

While these DCD-based improvements to PLR and PAIRED empirically lead to more robust policies, it is important to emphasize that our theoretical results only prove a minimax regret guarantee at NE for these methods; however, they provide no explicit guarantee of convergence to such NE. Further, it is worth highlighting that replay-based methods like $PLR^{\perp}$ are completely dependent on the quality of levels proposed by the generator. Our results show that simply curating high regret levels discovered via random search is enough to outperform the RL-based PAIRED teacher in the domains studied. We expect that advancing methods for defining or adapting the generator's proposal distribution holds great potential to improve the efficacy of our methods, especially in more complex, higher-dimensional domains, where random search may prove ineffective for finding useful training levels. Importantly, our methods assume an appropriate choice of what constitutes the UPOMDP's free parameters. Our methods cannot be expected to produce robust policies for zero-shot transfer if the set of environments defined by the free parameters does not sufficiently align with the transfer domain of interest. Designing the environment parameterization for successful zero-shot transfer to a specific target domain can be highly non-trivial, posing an important problem for future research.

Looking beyond environment design, we notice that long-running UED processes in expansive UPDOMPs closely resemble continual learning in open-ended domains. The congruency of these settings suggests our contributions around DCD may extend to more general continual learning problems in which agents must learn to master a diverse sequence of tasks with predefined (or inferred) episode boundaries—if tasks are assumed to be designed by a regret-maximizing teacher. Thus, DCD-based methods like $PLR^{\perp}$ may yield more general policies for continual learning. We anticipate many exciting crossovers between these areas of research in the years to come.

Given the rapid progress in applying RL to ever more complex domains, we can confidently expect a continued rise in real-world deployments of RL systems in the coming years. Unlike in simulation, in the real world, the environment tends to exhibit much more variability, which may not be explicitly coded into the associated simulator used for training. Deployed RL agents are thus liable to make many mistakes due to unexpected environment variations. Our methods for improving UED lead to more robust RL agents across a potentially wide range of changes to the environment. Thus, our work may prove to be a useful tool in attaining safer, more reliable RL agents, helping to enable the application of RL to more real-world problems.

By increasing the applicability of RL to real-world settings, our work may exacerbate the more general risks of deploying machine learning: increased unemployment; overreliance on biased models that potentially reinforce common misconceptions and societal inequalities; and the advancement of automated weapons. Particular to our methods, as discussed in Section 8, aligning the choice of free parameters for the UPOMDP to the target domain of interest is important for successful transfer. While this choice of free parameters then acts as a potential point of failure, the UPOMDP abstraction underlying UED reveals this problem generally impacts all RL methods aiming to train robust policies; a UPOMDP simply makes explicit the otherwise implicit space of environment configurations defined by a standard POMDP. By forcing us to consider where our training environment departs from reality, UED methods encourage designing RL systems in a way that is more aware of the underlying assumptions about the environment, thereby leading to more principled, robust systems.

## Acknowledgments and Disclosure of Funding

We would like to thank Natasha Jaques, Patrick Labatut, and Heinrich Küttler for fruitful discussions that helped inform this work. Further, we are grateful to our anonymous reviewers for their valuable feedback. MJ is supported by the FAIR PhD program. This work was funded by Facebook.

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
