# A  Theoretical Results

In this section we prove the theoretical results around the dual curriculum game and use these results to show approximation bounds for our methods, given that they have reached a Nash equilibrium (NE).

The first theorem is the main result that allows us to analyze dual curriculum games. The high-level result says that the NE of a dual curriculum game are approximate NE of the base game from the perspective of any of the individual players, or from the perspective of the joint strategy.

**Theorem 1.** *Let $B$ be the maximum difference between $U_t^1$ and $U_t^2$, and let $(\pi, \theta^1, \theta^2)$ be a NE for $\overline{G}$. Then $(\pi, p\theta^1 + (1-p)\theta^2)$ is an approximate NE for the base game with either teacher or for a teacher optimizing their joint objective. More precisely, it is a $2Bp(1-p)$-approximate NE when $U_t = pU_t^1 + (1-p)U_t^2$, a $2B(1-p)$-approximate NE when $U_t = U_t^1$, and a $2Bp$-approximate NE when $U_t = U_t^2$.*

At a high level, this is true because, for low values of $p$, the best-response strategies for the individual players can be thought of as approximate-best response strategies for the joint-player, and vis-versa. Since the Nash Equilibrium consists of each of the players playing their own best response, they must be playing an approximate best response for the joint-player. We provide a formal proof below:

*Proof.* Let $B$ be the maximum difference between $U_t^1$ and $U_t^2$, and let $(\pi, \theta^1, \theta^2)$ be a Nash Equilibrium for $\overline{G}$. Then consider $p\theta^1 + (1-p)\theta^2$ as a strategy in the base game for the joint player $pU_t^1 + (1-p)U_t^2$. Let $\theta^{1+2}$ be the best response for the joint player to $\pi$. Since $\pi$ is a best response by assumption, it is sufficient to show that $p\theta^1 + (1-p)\theta^2$ is an approximate best response. We then have

$$U_t(\pi, p\theta^1 + (1-p)\theta^2) \tag{6}$$
$$=p^2 U_t^1(\pi, \theta^1) + p(1-p)U_t^2(\pi, \theta^1) + p(1-p)U_t^1(\pi, \theta^2) + (1-p)^2 U_t^2(\pi, \theta^2) \tag{7}$$
$$\geq p^2 U_t^1(\pi, \theta^1) + p(1-p)(U_t^1(\pi, \theta^1) - B) + p(1-p)(U_t^2(\pi, \theta^2) - B) + (1-p)^2 U_t^2(\pi, \theta^2) \tag{8}$$
$$=pU_t^1(\pi, \theta^1) + (1-p)U_t^2(\pi, \theta^2) - 2Bp(1-p) \tag{9}$$
$$\geq U_t(\pi, \theta^{1+2}) - 2Bp(1-p) \tag{10}$$

Thus, we have shown that $(\pi, p\theta^1 + (1-p)\theta^2)$ represents an $2Bp(1-p)$-Nash equilibrium for the joint player. For the first teacher we have the opposite condition trivially, the teacher is doing a best response to the student. We must now show that the student is doing an approximate best response to the teacher.

Let $\pi^1$ be the best response to the first teacher (with utility $U_t^1$) and let $\pi^{1+2}$ be the best response policy to the joint teacher. In this argument we will start with the observation that $U_s(\pi^1, \theta^{1+2}) \leq U_s(\pi^{1+2}, \theta^{1+2})$ by definition, and then argue that we can construct an upper bound on the performance of $\pi^1$ on $\theta^1$, $U_s(\pi^1, \theta^1)$, and a lower bound on the performance of $\pi^{1+2}$ on $\theta^1$, $U_s(\pi^{1+2}, \theta^1)$. We get the desired result by combining these two arguments.

First we use $U_s(\pi^1, \theta^{1+2})$ to upper bound $U_s(\pi^1, \theta^1)$:

$$U_s(\pi^1, \theta^{1+2}) = pU_s(\pi^1, \theta^1) + (1-p)U_s(\pi^1, \theta^2) \tag{11}$$
$$\geq pU_s(\pi^1, \theta^1) + (1-p)(U_s(\pi^1, \theta^1) - B) \tag{12}$$
$$= U_s(\pi^1, \theta^1) - (1-p)B \tag{13}$$

Second we can use $U_s(\pi^{1+2}, \theta^{1+2})$ to lower bound $U_s(\pi^{1+2}, \theta^1)$:

$$U_s(\pi^{1+2}, \theta^{1+2}) = pU_s(\pi^{1+2}, \theta^1) + (1-p)U_s(\pi^{1+2}, \theta^2) \tag{14}$$
$$\leq pU_s(\pi^{1+2}, \theta^1) + (1-p)(U_s(\pi^{1+2}, \theta^1) + B) \tag{15}$$
$$= U_s(\pi^{1+2}, \theta^1) + (1-p)B \tag{16}$$

Putting this all together, we have

$$U_s(\pi^{1+2}, \theta^1) + (1-p)B \geq U_s(\pi^1, \theta^1) - (1-p)B.$$

Which, after rearranging terms, gives

$$U_s(\pi^{1+2}, \theta^1) \geq U_s(\pi^1, \theta^1) - 2(1-p)B$$

as desired. Repeating the symmetric argument shows the desired property for the second teacher. □

Following this main theorem, we can apply it to two of our methods. First we can apply it to naive PLR, which trains on a mixture of domain randomization (a teacher with utility $U_t^C$) and the PLR bandit (a teacher with utility $U_t^R$). This result shows that as we reduce the number of random episodes, the approximation to a minimax regret strategy improves. The intuition behind this is a direct application of Theorem 1, to show that it is an approximate Nash for the minimax regret player, and then showing that the minimax reget player has access to a strategy which ensures small regret, thus the regret that the equilibrium ensures must be approximately small.

**Corollary 1.** *Let $\overline{G}$ be the dual curriculum game in which the first teacher maximizes regret, so $U_t^1 = U_t^R$, and the second teacher plays randomly, so $U_t^2 = U_t^U$. Let $V^\theta(\pi)$ be bounded in $[B^-, B^+]$ for all $\theta, \pi$. Further, suppose that $(\pi, \theta^1, \theta^2)$ is a Nash equilibrium of $\overline{G}$. Let $R^* = \min_{\pi_A \in \Pi}\{\max_{\theta, \pi_B \in \Theta, \Pi}\{\text{REGRET}^\theta(\pi_A, \pi_B)\}\}$ be the optimal worst-case regret. Then $\pi$ is $2(B^+ - B^-)(1 - p)$ close to having optimal worst-case regret, or formally, $\max_{\theta, \pi_B \in \Theta, \Pi}\{\text{REGRET}^\theta(\pi_A, \pi)\} \geq R^* - 2(B^+ - B^-)(1-p)$. Moreover, there exists environments for all values of $p$ within a constant factor of achieving this bound.*

*Proof.* Since $V^\theta(\pi)$ is bounded in $[B^-, B^+]$ for all $\theta, \pi$, we know that $U_t^1$ and $U_t^2$ are within $(B^+ - B^-)$ of each other. Thus by Theorem 1 we have that $(\pi, \theta^1, \theta^2)$ is a $2(B^+ - B^-)(1 - p)$-Nash equilibrium of the base game when $U_t = U_t^1$. Thus $\pi$ is a $2(B^+ - B^-)(1 - p)$ approximate best-response to $\theta^1$. However, since $\theta^1$ is a best response it chooses a regret maximizing parameter distribution. Thus the $2(B^+ - B^-)(1 - p)$ does not just measure the sub-optimally of $\pi$ with respect to $\theta^1$, but measures the worst-case regret of $\pi$ across all $\theta$ as desired.

The intuition for the existence of examples in which this approximation of regret decays linearly in $p$ is that a random level and the maximal regret level can be very different, and so the two measures may diverge drastically. For an example environment where $\pi$ deviates strongly from the minimax regret strategy, consider the one-step UMDP described in Table 1.

|         | $\theta_0$      | $\theta_1$      | $\theta_2 \ldots \theta_n$  |
|---------|-----------------|-----------------|-----------------------------|
| $\pi_0$ | $B$             | $0$             | $0$                         |
| $\pi_1$ | $0$             | $B$             | $0$                         |
| $\pi_2$ | $Bp + 2\epsilon$| $0$             | $\frac{Bp}{2} + \epsilon$   |
| $\pi_3$ | $0$             | $Bp + 2\epsilon$| $\frac{Bp}{2} + \epsilon$   |

Table 1: In this environment all payoffs are between 0 and $B$(for $p \in (0, 1)$ and $\epsilon < \frac{B(1-p)}{2}$), where $B$ is assumed to be positive. Randomizing between $\pi_0$ and $\pi_1$ minimizes regret, but choosing $\pi_2$ or $\pi_3$ is better in expectation under the uniform distribution. For large $n$ it is especially clear that $\pi_2$ and $\pi_3$ have better expected value under the uniform distribution, though we show that even for $n = 2$, the optimal joint policy can mix between $\pi_2$ and $\pi_3$ incurring high regret.

Note that in Table 1, no policy has less than $\frac{B}{2}$ regret, since every policy will have to incur $B$ regret on either $\{\theta_0, \theta_1\}$ at least half the time. The minimax regret policy mixes uniformly between $\pi_0$ and $\pi_1$ to achieve regret of exactly $\frac{B}{2}$. We can ignore $\theta_2 \ldots \theta_n$ for the regret calculations by assuming that $\epsilon < \frac{B(1-p)}{2}$, since every policy achieves less than $\frac{B}{2}$ regret on these levels.

Our claim is that in equilibrium of $\overline{G}$ in this environment, the student policy can incur $\frac{B}{2} + \frac{B(1-p)}{2} - \epsilon$ regret, $\frac{B(1-p)}{2} - \epsilon$ more than the minimax regret policy. An example of such an equilibrium point would be when the student policy uniformly randomizes between $\pi_2$ and $\pi_3$, which we will call $\pi_{2+3}$, when the minimax teacher uniformly randomizes between $\theta_0$ and $\theta_1$ which we will call $\theta_{0+1}$, and

when the uniform teacher randomizes exactly which we call $\tilde{\theta}$. To check this we must show that $(\pi_{2+3}, \theta_{0+1}, \tilde{\theta})$ is in fact a NE of $\overline{G}$. Then we must show that $\pi_{2+3}$ incurs $\frac{B}{2} + \frac{B(1-p)}{2} - \epsilon$ regret.

To show that $(\pi_{2+3}, \theta_{0+1}, \tilde{\theta})$ is a NE of $\overline{G}$ first note that $\tilde{\theta}$ is trivially a best response for the uniform utility function. Also note that $\theta_{0+1}$ maximizes the regret of $\pi_{2+3}$ since $\theta_0$ and $\theta_1$ are the only two parameters on which $\pi_{2+3}$ incur regret, and they incur the same regret; thus, any mixture over them will be optimal for the regret-based teacher. Finally, we need to show that $\pi_{2+3}$ is optimal for the student. To do this we will calculate the expected value of each policy and notice that the expected values for $\pi_2$ and $\pi_3$ are higher than for $\pi_0$ and $\pi_1$. Thus any optimal policy will place no weight on $\pi_0$ and $\pi_1$, but any distribution over $\pi_2$ and $\pi_3$ will be equivalently optimal. By symmetry, we can show only the calculations for $\pi_0$ and $\pi_2$:

$$\pi_0 = p(\frac{1}{2}B + \frac{1}{2}0) + (1-p)0 = \frac{Bp}{2} \tag{17}$$

$$\pi_2 = p(\frac{1}{2}(Bp + 2\epsilon) + \frac{1}{2}0) + (1-p)(\frac{Bp}{2} + \epsilon) = \frac{Bp}{2} + \epsilon \tag{18}$$

Thus $\pi_2$ and $\pi_3$ achieve $\epsilon$ higher expected value by the joint distribution. Thus, we know that $\pi_{2+3}$ is a best response and $(\pi_{2+3}, \theta_{0+1}, \tilde{\theta})$ is in fact a NE of $\overline{G}$.

Finally, we simply need to show that $\pi_{2+3}$ incurs $\frac{B}{2} + \frac{B(1-p)}{2} - \epsilon$ regret. WLOG, we can evaluate its regret on $\theta_0$. On $\theta_0$, $\pi_{2+3}$ achieves $\frac{Bp}{2} + \epsilon$ reward while $\pi_0$ achieves $B$. Thus $\pi_{2+3}$ incurs regret of $B - (\frac{Bp}{2} + \epsilon) = \frac{B}{2} + \frac{B-Bp}{2} - \epsilon = \frac{B}{2} + \frac{B(1-p)}{2} - \epsilon$ as desired. As discussed before, since the minimax regret policy achieves $\frac{B}{2}$, this is $\frac{B(1-p)}{2} - \epsilon$ more regret than optimal. $\square$

Lastly, we can also apply Theorem 1 to prove that REPAIRED achieves a minimax regret strategy in equilibrium. The intuition behind this corollary is that, since the utility functions of both teachers are the same, the approximate NE ensured by Theorem 1 is actually a true NE; therefore, the minimax theorem applies.

**Corollary 2.** *Let $\overline{G}$ be the dual curriculum game in which both teachers maximize regret, so $U_t^1 = U_t^2 = U_t^R$. Further, suppose that $(\pi, \theta^1, \theta^2)$ is a Nash equilibrium of $\overline{G}$. Then, $\pi \in \text{argmin}_{\pi_A \in \Pi}\{\max_{\theta, \pi_B \in \Theta, \Pi}\{\text{REGRET}^\theta(\pi_A, \pi_B)\}\}$.*

*Proof.* Since $U_t^1 = U_t^2 = U_t^R$ the joint objective is $pU_t^1 + (1-p)U_t^2 = U_t^R$. Note that since $U_t^1 = U_t^2$, $B = 0$. Thus by Theorem 1 $(\pi, p\theta^1 + (1-p)\theta^2)$ is a 0-Nash Equilibrium of the base game with teacher objective $U_t^R$, thus by the minimax theorem, $\pi \in \text{argmin}_{\pi_A \in \Pi}\{\max_{\theta, \pi_B \in \Theta, \Pi}\{\text{REGRET}^\theta(\pi_A, \pi_B)\}\}$ as desired. $\square$

# B Algorithms

Although the PLR update rule for the level buffer of size $K$ in the case of unbounded training levels is described in [17], we provide the pseudocode for this update rule in Algorithm 2 for completeness. Given staleness coefficient $\rho$, temperature $\beta$, a prioritization function $h$ (e.g. rank), level buffer scores $S$, level buffer timestamps $C$, and the current episode count $c$ (i.e. current timestamp), the $P_{\text{replay}}$ update takes the form

$$P_{\text{replay}} = (1 - \rho) \cdot P_S + \rho \cdot P_C,$$
$$P_S = \frac{h(S_i)^{1/\beta}}{\sum_j h(S_j)^{1/\beta}},$$
$$P_C = \frac{c - C_i}{\sum_{C_j \in C} c - C_j}.$$

The pseudocode for Replay-Enhanced PAIRED (REPAIRED), the method described in Section 5, is presented in Algorithm 3.

---

**Algorithm 2:** PLR level-buffer update rule

---

**Input:** Level buffer $\boldsymbol{\Lambda}$ of size $K$ with scores $S$ and timestamps $C$; level $\theta$; level score $S_\theta$; and current episode count $c$

**if** $|\boldsymbol{\Lambda}| < K$ **then**
    Insert $\theta$ into $\boldsymbol{\Lambda}$, and set $S(\theta) = S_\theta$, $C(\theta) = c$
**else**
    Find level with minimal support, $\theta_{\min} = \arg\min_\theta P_{\text{replay}}(\theta)$

    **if** $S(\theta_{\min}) < S_\theta$ **then**
        Remove $\theta_{\min}$ from $\boldsymbol{\Lambda}$
        Insert $\theta$ into $\boldsymbol{\Lambda}$, and set $S(\theta) = S_\theta$, $C(\theta) = c$
        Update $P_{\text{replay}}$ with latest scores $S$ and timestamps $C$
    **end**
**end**

---

**Algorithm 3:** REPAIRED

---

Randomly initialize Protagonist, Antagonist, and Generator policies $\pi^A(\phi^A)$, $\pi^B(\phi^B)$, and $\tilde{\theta}$
Initialize Protagonist and Antagonist PLR level buffers $\boldsymbol{\Lambda}^A$ and $\boldsymbol{\Lambda}^B$
**while** *not converged* **do**
    Sample replay-decision Bernoulli, $d \sim P_D(d)$
    **if** $d = 0$ **then**
        Teacher policy $\tilde{\theta}$ generates the next level, $\theta$
        Set $\theta^A = \theta^B = \theta$
        Collect trajectory $\tau^A$ on $\theta^A$ and $\tau^B$ on $\theta^B$ with stop-gradients $\phi_\perp^A$, $\phi_\perp^B$
        Update $\tilde{\theta}$ with $\text{REGRET}^\theta(\pi^A, \pi^B)$
    **else**
        PLR samples replay levels, $\theta^A \sim \boldsymbol{\Lambda}^A$ and $\theta^B \sim \boldsymbol{\Lambda}^B$
        Collect trajectory $\tau^A$ on $\theta^A$ and $\tau^B$ on $\theta^B$
        Update $\pi^A$ with rewards $\boldsymbol{R}(\tau^A)$, and $\pi^B$, with rewards $\boldsymbol{R}(\tau^B)$
    **end**
    Compute PLR score $S^A = \textbf{score}(\tau^A, \tau^B, \pi^A)$
    Compute PLR score $S^B = \textbf{score}(\tau^B, \tau^A, \pi^B)$
    Update $\boldsymbol{\Lambda}^A$ with $\theta^A$ using score $S^A$
    Update $\boldsymbol{\Lambda}^B$ with $\theta^B$ using score $S^B$
**end**

---

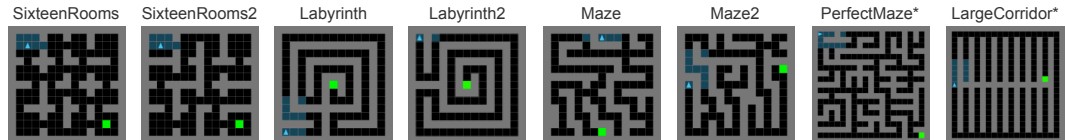

Figure 7: Test maze environments for evaluating zero-shot transfer. An asterisk (*) next to the maze name indicates the maze is procedurally-generated, and thus each attempt corresponds to a random configuration of the maze.

## C   Additional Experimental Results

This section provides additional experimental results in MiniGrid and CarRacing environments. Note that we determine the statistical significance of our results using a Welch t-test [43].

### C.1   Extended Results for MiniGrid

Unlike the original maze experiments used to evaluate PAIRED in [10], we conduct our main maze experiments with a block budget of 25 blocks (reported in Section 6.1), rather than 50 blocks. Following the environment parameterization in [10], for a block budget of $B$, the teacher attempts to place $B$ blocks that act as obstacles when designing each maze level. However, the teacher can place fewer than $B$ blocks, as placing a block in a location already occupied by a block results in a no-opt. We found that PAIRED underperforms DR when both methods are given a budget of 50 blocks, a setting in which randomly sampled mazes exhibit enough structural complexity to allow DR to learn highly robust policies. Note that [10] used a DR baseline with a 25-block budget. With a 50-block budget, DR and all replay-based methods are able to fully solve almost all test mazes after around 500M steps of training, making UED of mazes with a 50-block budget too simple of a setting to provide an informative comparison among the methods studied.

### C.1.1   Mazes with a 25-block budget

We report the results of evaluating policies produced by each method after 250M training steps on each of the zero-shot transfer environments in Figure 8 and Table 2. Examples of each test environment are presented in Figure 7. All replay-based UED methods lead to policies with statistically significantly ($p < 0.05$) higher test performance than PAIRED, and PLR$^{\perp}$, after 500M training steps, similarly improves over PLR when trained for an equivalent number of gradient updates (as replay rate is set to 0.5). Note that for PAIRED and REPAIRED, we evaluate the protagonist policy.

To provide a further sense of the training dynamics, we present the per-agent training returns for each method in Figure 9. Notably PAIRED results in antagonists that attain higher returns than the protagonist as expected. This dynamic takes on a mild oscillation, visible in the training return curve of the generator (adversary). As the protagonist adapts to the adversarial levels, the generator's return reduces, until the generator discovers new configurations that better exploit the relative differences between the two student policies. Notably, the adversary under REPAIRED seems to propose more difficult levels for both the protagonist and antagonist, while the resulting protagonist policy exhibits improved test performance, as seen in Figure 4.

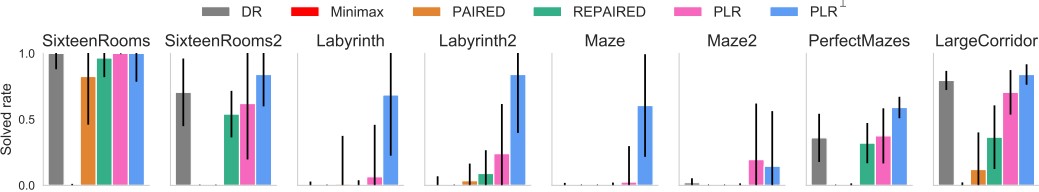

Figure 8: Zero-shot test performance on OOD environments when trained with a 25-block budget. The plots report the median and interquartile range of solved rates over 10 runs.

Table 2: Mean test returns and standard errors on zero-shot transfer mazes for each method using a 25-block budget. Results are aggregated over 100 attempts for each maze across 10 runs per method. Bolded figures overlap in standard error with the method attaining the maximum mean test return in each row.

| Environment | DR | Minimax | PAIRED | REPAIRED | PLR | PLR$^\perp$ | PLR$^\perp$ (500M) |
|---|---|---|---|---|---|---|---|
| Labyrinth | $0.2 \pm 0.1$ | $0.0 \pm 0.0$ | $0.3 \pm 0.1$ | $0.1 \pm 0.0$ | $0.3 \pm 0.1$ | $\mathbf{0.5 \pm 0.1}$ | $\mathbf{0.7 \pm 0.1}$ |
| Labyrinth2 | $0.2 \pm 0.1$ | $0.0 \pm 0.0$ | $0.2 \pm 0.1$ | $0.2 \pm 0.1$ | $0.4 \pm 0.1$ | $\mathbf{0.6 \pm 0.1}$ | $\mathbf{0.8 \pm 0.1}$ |
| LargeCorridor | $\mathbf{0.7 \pm 0.1}$ | $0.1 \pm 0.1$ | $0.3 \pm 0.1$ | $0.5 \pm 0.1$ | $\mathbf{0.7 \pm 0.1}$ | $\mathbf{0.8 \pm 0.1}$ | $\mathbf{0.8 \pm 0.1}$ |
| Maze | $0.0 \pm 0.0$ | $0.0 \pm 0.0$ | $0.0 \pm 0.0$ | $0.2 \pm 0.1$ | $0.3 \pm 0.1$ | $\mathbf{0.6 \pm 0.1}$ | $0.5 \pm 0.1$ |
| Maze2 | $0.0 \pm 0.0$ | $0.0 \pm 0.0$ | $0.1 \pm 0.1$ | $0.1 \pm 0.1$ | $\mathbf{0.4 \pm 0.1}$ | $\mathbf{0.4 \pm 0.1}$ | $\mathbf{0.5 \pm 0.1}$ |
| PerfectMaze | $0.3 \pm 0.1$ | $0.0 \pm 0.0$ | $0.0 \pm 0.0$ | $0.4 \pm 0.1$ | $0.4 \pm 0.1$ | $\mathbf{0.6 \pm 0.1}$ | $0.5 \pm 0.1$ |
| SixteenRooms | $0.9 \pm 0.0$ | $0.1 \pm 0.1$ | $0.7 \pm 0.1$ | $0.9 \pm 0.1$ | $\mathbf{1.0 \pm 0.0}$ | $0.8 \pm 0.1$ | $\mathbf{1.0 \pm 0.0}$ |
| SixteenRooms2 | $\mathbf{0.7 \pm 0.1}$ | $0.0 \pm 0.0$ | $0.0 \pm 0.0$ | $\mathbf{0.6 \pm 0.1}$ | $0.5 \pm 0.1$ | $\mathbf{0.7 \pm 0.1}$ | $\mathbf{0.7 \pm 0.1}$ |
| Mean | $0.4 \pm 0.0$ | $0.0 \pm 0.0$ | $0.2 \pm 0.0$ | $0.4 \pm 0.0$ | $0.5 \pm 0.1$ | $\mathbf{0.6 \pm 0.1}$ | $\mathbf{0.7 \pm 0.1}$ |

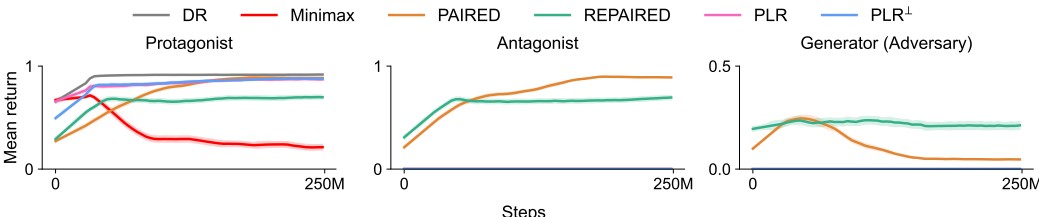

Figure 9: Training returns for each participating agent in each method, when trained with a 25-block budget. Plots show the mean and standard error over 10 runs.

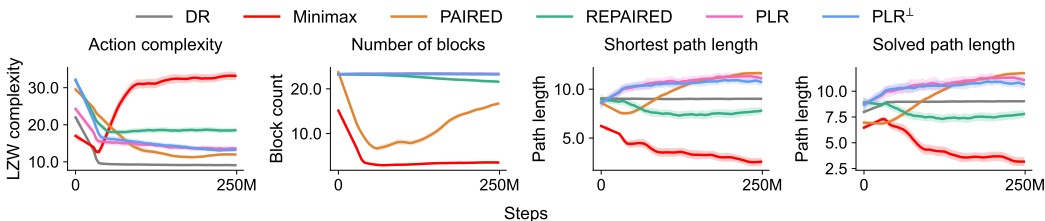

Figure 10: Complexity metrics of environments generated by the teacher throughout training with a 25-block budget. Plots show the mean and standard error of 10 runs.

Additional complexity metrics tracked during training are shown in Figure 10. Alongside the number of blocks and shortest path length of levels seen during training, we also track solved path length and action complexity. Solved path length corresponds to the shortest path length from start position to goal in the levels successfully solved by the primary student agent (e.g. the protagonist in PAIRED). Action complexity corresponds to the Lempel-Ziv-Welch (LZW) complexity—a commonly used measure of string compressibility—of the action sequence taken during the primary student agent's trajectories. As expected, DR results in constant complexity for number of blocks and path length metrics. REPAIRED generates mazes with significantly greater complexity in terms of block count. The lower path lengths seen by REPAIRED suggest that it trains agents that more readily generalize to different path lengths, thereby pressuring the adversary to raise complexity in terms of block count. Further, given the high replay rates used, the REPAIRED adversary sees far fewer gradient updates with which to adjust its policy. As its shortest path lengths exceed that of PAIRED after adjusting proportionately by replay rate, foreseeably, over a longer period, the shortest path lengths generated by REPAIRED may meet or exceed that of PAIRED. In all cases, the action complexity reduces as the agent becomes more decisive, and we see that both PAIRED and REPAIRED lead to more decisive policies—as indicated by the simultaneously lower action complexity and greater level complexity in terms of higher block count (relative to DR) and, in the case of PAIRED, higher path length metrics. Lastly, it is interesting to note that while the random generator used by PLR produces levels of average complexity, the complexity of curated levels, as revealed in Figure 4, is significantly higher and, in the case of path length, steadily increasing.

### C.1.2 Mazes with a 50-block budget

Similarly, Figures 12, 13, and 14 report the training dynamics and test performance of agents trained using each method with a 50-block budget for 500M steps. Figure 11 shows that DR and all replay-based methods are able to reach near perfect solve rates on most test mazes after 500M steps of training, with the exception of the Maze and PerfectMaze environments, where the test performances across methods are not markedly dissimilar, making the setting with a 50-block budget uninformative for assessing performance differences among these methods. The example mazes generated by each method, presented in Figure 15, shows that the larger block budget allows DR to sample mazes with greater structural complexity, leading to robust policies and diminishing the benefits of the UED methods studied. Therefore, in this work, we focus the main results for the maze domain on the more challenging setting with a 25-block budget. Note that the impact of the block budget on test performance further highlights the importance of properly adapting the training distribution for producing policies exhibiting high generality—a problem that our replay-based UED methods effectively address, as demonstrated by the results for the 25-block setting.

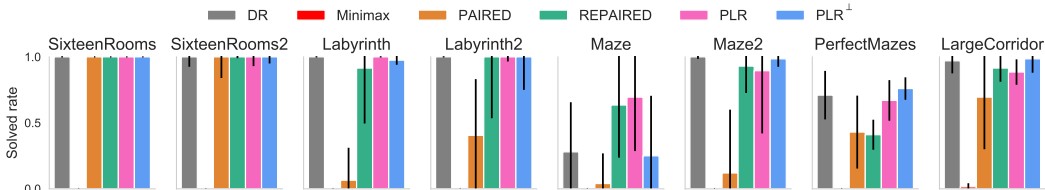

Figure 11: Zero-shot test performance on OOD environments when trained with a 50-block budget. The plots show the median and interquartile range of solved rates over 10 runs.

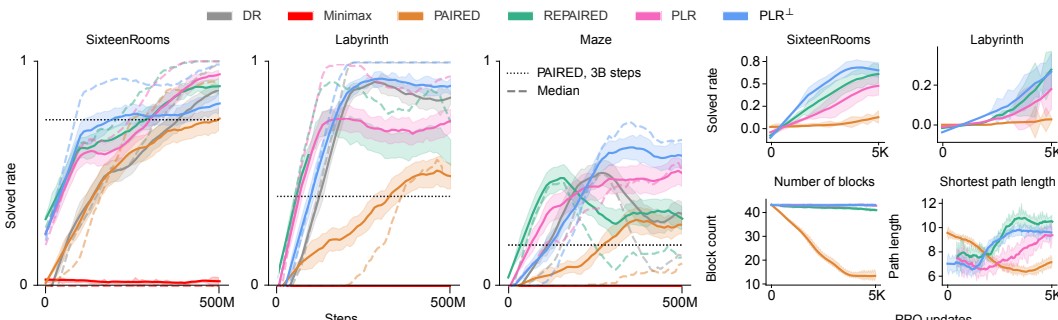

Figure 12: Test performance as a function of number of training steps with a 50-block budget (left), and test performance and complexity metrics as a function of number of PPO updates (right). The plots show the mean and standard error over 10 runs.

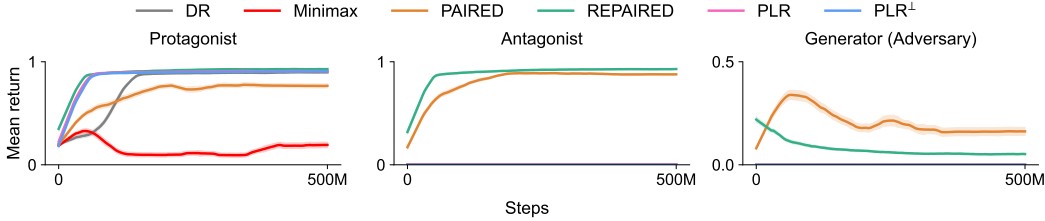

Figure 13: Training returns for each participating agent in each method when training with a 50-block budget. Plots show the mean and standard error over 10 runs.

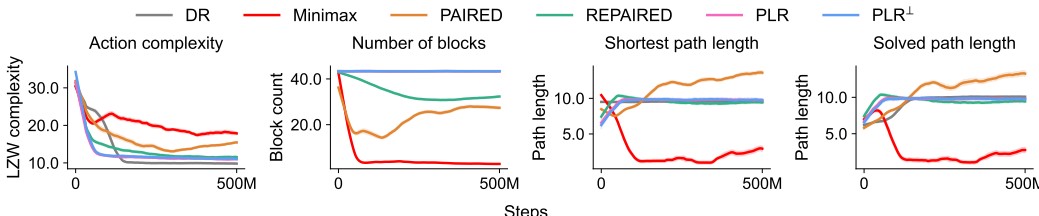

Figure 14: Complexity metrics of environments generated by the teacher throughout training with a 50-block budget. Plots show the mean and standard error of 10 runs.

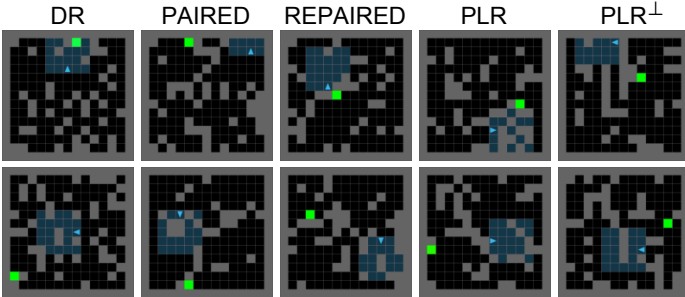

Figure 15: Example mazes generated by each method when using a 50-block budget.

## C.2 Extended Results for CarRacing

The training return plots for each agent, shown in Figure 16, reveal that PAIRED's generator (adversary) overexploits the relative advantages of the antagonist over the protagonist, leading to a highly suboptimal protagonist policy. In fact, as shown in the right-most plot of Figure 16, the resulting protagonist policies suffer such performance degradation from the adversarial curriculum that they can no longer even successfully drive on the original, simpler CarRacing tracks.

Additionally, we present per-track zero-shot transfer returns for the entire CarRacing-F1 benchmark after 5M training steps (equivalent to 40M environment interaction steps due to the usage of action repeat) in Table 3. Results report the mean and standard deviation over 100 attempts per track across 10 seeds. While DR acts as a strong baseline in terms of zero-shot generalization in this setting, $PLR^\perp$ either attains the highest mean return, or matches the method achieving the highest return within standard error on all tracks. The mean performance of $PLR^\perp$ across the full benchmark is statistically significantly higher ($p < 0.001$) than that of all other methods. Notably, PAIRED sees poor results, likely due to the generator's ability to overexploit the differences between antagonist and protagonist to detrimental effect in this domain. We see that REPAIRED mitigates this effect to a degree, resulting in more competitive policies. Note that due to the high compute overhead of training the AttentionAgent (8.2 billion steps of training over a population 256 agents) [38], we resorted to evaluating its mean F1 performance using the pre-trained model weights provided by the authors with their public code release. As a result, we only have a single training run for AttentionAgent. This means we cannot reliably compute standard errors for this baseline, but we believe that showing the performance for a single training seed of AttentionAgent on the F1 benchmark alongside our methods, as done in Figure 6, nonetheless provides a useful comparison for further contextualizing the efficacy of our methods. This comparison highlights how, by only modifying the training curriculum, our methods produce policies with test returns exceeding that of AttentionAgent—which in contrast, uses a powerful attention-based policy and a much larger number of training steps.

As a further analysis of robustness, we inspect the minimum returns over 10 attempts per track, averaged over 10 runs per method. We present these results (mean and standard error) in Figure 17. $PLR^\perp$ achieves consistently higher minimum returns on average for many of the tracks compared to the other methods, including on the challenging Russia and USA tracks. The fact that simply curating random levels, as done by $PLR^\perp$, more reliably approaches a minimax regret policy than PAIRED and REPAIRED suggests that RL may not be an effective means for optimizing the PAIRED teacher.

Table 3: Mean test returns and standard errors of each method on the full F1 benchmark. Results are aggregated over 10 attempts for each track across 10 runs per method. Bolded figures overlap in standard error with the method attaining the maximum mean test return in each row. We see that PLR$^\perp$ consistently either outperforms the other methods or matches PLR, the next best performing method. Note that we separately report the results of a single run for AttentionAgent due to its high compute overhead.

| Track | DR | PAIRED | REPAIRED | PLR | PLR$^\perp$ | AttentionAgent |
|---|---|---|---|---|---|---|
| Australia | $484 \pm 29$ | $100 \pm 22$ | $414 \pm 27$ | $545 \pm 23$ | $\mathbf{692 \pm 15}$ | 826 |
| Austria | $409 \pm 21$ | $92 \pm 24$ | $345 \pm 19$ | $442 \pm 18$ | $\mathbf{615 \pm 13}$ | 511 |
| Bahrain | $298 \pm 27$ | $-35 \pm 19$ | $295 \pm 23$ | $411 \pm 22$ | $\mathbf{590 \pm 15}$ | 372 |
| Belgium | $328 \pm 16$ | $72 \pm 20$ | $293 \pm 19$ | $327 \pm 15$ | $\mathbf{474 \pm 12}$ | 668 |
| Brazil | $309 \pm 23$ | $76 \pm 18$ | $256 \pm 19$ | $387 \pm 17$ | $\mathbf{455 \pm 13}$ | 145 |
| China | $115 \pm 24$ | $-101 \pm 9$ | $7 \pm 18$ | $84 \pm 20$ | $\mathbf{228 \pm 24}$ | 344 |
| France | $279 \pm 32$ | $-81 \pm 13$ | $240 \pm 29$ | $290 \pm 35$ | $\mathbf{478 \pm 22}$ | 153 |
| Germany | $274 \pm 23$ | $-33 \pm 16$ | $272 \pm 22$ | $388 \pm 20$ | $\mathbf{499 \pm 18}$ | 214 |
| Hungary | $465 \pm 32$ | $98 \pm 29$ | $414 \pm 29$ | $533 \pm 26$ | $\mathbf{708 \pm 17}$ | 769 |
| Italy | $461 \pm 27$ | $132 \pm 24$ | $371 \pm 25$ | $588 \pm 20$ | $\mathbf{625 \pm 12}$ | 798 |
| Malaysia | $236 \pm 25$ | $-26 \pm 17$ | $200 \pm 17$ | $283 \pm 20$ | $\mathbf{400 \pm 18}$ | 300 |
| Mexico | $458 \pm 33$ | $67 \pm 31$ | $415 \pm 30$ | $561 \pm 21$ | $\mathbf{712 \pm 12}$ | 580 |
| Monaco | $268 \pm 28$ | $-28 \pm 18$ | $256 \pm 26$ | $360 \pm 32$ | $\mathbf{486 \pm 19}$ | 835 |
| Netherlands | $328 \pm 26$ | $70 \pm 20$ | $307 \pm 21$ | $\mathbf{418 \pm 21}$ | $\mathbf{419 \pm 25}$ | 131 |
| Portugal | $324 \pm 27$ | $-49 \pm 13$ | $265 \pm 21$ | $407 \pm 15$ | $\mathbf{483 \pm 13}$ | 606 |
| Russia | $382 \pm 30$ | $51 \pm 21$ | $419 \pm 25$ | $479 \pm 24$ | $\mathbf{649 \pm 14}$ | 732 |
| Singapore | $336 \pm 29$ | $-35 \pm 14$ | $274 \pm 21$ | $386 \pm 22$ | $\mathbf{566 \pm 15}$ | 276 |
| Spain | $433 \pm 24$ | $134 \pm 24$ | $358 \pm 24$ | $482 \pm 17$ | $\mathbf{622 \pm 14}$ | 759 |
| UK | $393 \pm 28$ | $138 \pm 25$ | $380 \pm 22$ | $456 \pm 16$ | $\mathbf{538 \pm 17}$ | 729 |
| USA | $263 \pm 31$ | $-119 \pm 11$ | $120 \pm 25$ | $243 \pm 28$ | $\mathbf{381 \pm 33}$ | -192 |
| Mean | $341 \pm 22$ | $19 \pm 15$ | $293 \pm 18$ | $408 \pm 12$ | $\mathbf{534 \pm 7}$ | 477 |

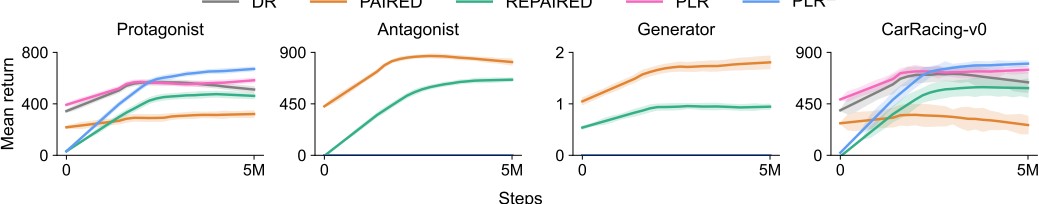

Figure 16: From left to right: Returns attained by the protagonist, antagonist, and generator (adversary) throughout training; the protagonist's zero-shot transfer performance on the original CarRacing-v0 during training. The mean and standard error over 10 runs are shown.

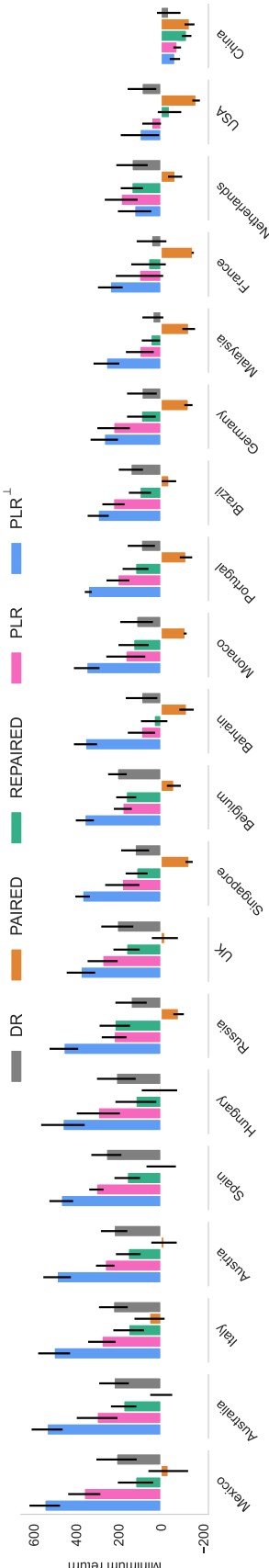

Figure 17: Minimum returns attained across 10 test episodes per track per seed. Bars report mean and standard error over 10 training runs.

Table 4: Hyperparameters used for training each method in the maze and car racing environments.

| Parameter | MiniGrid | CarRacing |
|---|---|---|
| *PPO* | | |
| $\gamma$ | 0.995 | 0.99 |
| $\lambda_{\text{GAE}}$ | 0.95 | 0.9 |
| PPO rollout length | 256 | 125 |
| PPO epochs | 5 | 8 |
| PPO minibatches per epoch | 1 | 4 |
| PPO clip range | 0.2 | 0.2 |
| PPO number of workers | 32 | 16 |
| Adam learning rate | 1e-4 | 3e-4 |
| Adam $\epsilon$ | 1e-5 | 1e-5 |
| PPO max gradient norm | 0.5 | 0.5 |
| PPO value clipping | yes | no |
| Return normalization | no | yes |
| Value loss coefficient | 0.5 | 0.5 |
| Student entropy coefficient | 0.0 | 0.0 |
| | | |
| *PLR* | | |
| Replay rate, $p$ | 0.5 | 0.5 |
| Buffer size, $K$ | 4000 | 8000 |
| Scoring function | MaxMC | positive value loss |
| Prioritization | rank | proportional |
| Temperature, $\beta$ | 0.3 | 1.0 |
| Staleness coefficient, $\rho$ | 0.3 | 0.7 |
| | | |
| $PLR^{\perp}$ | | |
| Scoring function | MaxMC | positive value loss |
| | | |
| *PAIRED* | | |
| Generator entropy coefficient | 0.0 | 0.0 |
| | | |
| *REPAIRED* | | |
| Generator entropy coefficient | 0.0 | 0.01 |
| Scoring function | MaxMC | MaxMC |

## D    Experiment Details and Hyperparameters

This section details the environments, agent architectures, and training procedures used in our experiments discussed in Section 6. We use PPO to train both student and generator policies in all experiments. Section 6 reports results for each method using the best hyperparameter settings, which we summarize in Figure 4. Note that unless specified, PPO hyperparameters are shared between student and teacher, and PLR hyperparameters are shared between REPAIRED and REPAIRED. The procedures for determining the hyperparameter choices for each environment are detailed below, in Sections D.1 and D.2.

### D.1    Partially-Observable Navigation (MiniGrid)

**Environment details** Our mazes are based on MiniGrid [7]. Each maze consists of a $15 \times 15$ grid, where each cell can contain a wall, the goal, the agent, or navigable space. The student agent receives a reward of $1 - T/T_{\max}$ upon reaching the goal, where $T$ is the episode length and $T_{\max}$ is the maximum episode length (set to 250). Otherwise, the agent receives a reward of 0 if it fails to reach the goal. The observation space consists of the agent's orientation (facing north, south, east, or west) and the $7 \times 7$ grid immediately in front of and including the agent. This grid takes the form of a 3-channel integer encoding. The action space consists of 7 total actions, though mazes only make use of the first three: turn left, turn right, and forward. We do not mask out irrelevant actions.

**Level generation** Each maze is fully surrounded by walls, resulting in $13 \times 13 = 169$ cells in which the generator can place walls, the goal, and the agent. Starting from an initially empty maze (except the bordering walls), the generator is given a budget of $W = 50$ steps in which it can choose a grid cell in which to place a wall. Placing a wall in a cell already containing a wall results in a no-opt.

After wall placement, the generator then chooses cells for the goal and the agent's starting position. If either of these cells collides with an existing wall, a random empty cell is chosen. At each time step, the generator teacher receives the full grid observation of the developing maze, the one-hot encoding of the current time step, as well as a 50-dimensional random noise vector, where each component is uniformly sampled from $[0.0, 1.0]$.

**Generator architecture** We base the generator architecture on the the original model used for the PAIRED adversary in [10]. This model encodes the full grid observation using a convolution layer ($3 \times 3$ kernel, stride length 1, 128 filters) followed by a ReLU activation layer over the flattened convolution outputs. The current time step is embedded into a 10-dimensional space, which is concatenated to the grid embedding, along with the random noise vector. This combined representation is then passed through an LSTM with hidden dimension 256, followed by two fully-connected layers, each with a hidden dimension 32 and ReLU activations, to produce the action logits over the 169 possible cell choices. We further ablated the LSTM and found that its absence preserves the performance of the minimax generator in both 25-block and 50-block settings, as well as that of the PAIRED generator in the 50-block setting, as expected given that the full grid and time step form a Markov state. However, the PAIRED generator struggles to learn without an LSTM in the 25-block setting. We believe PAIRED's improved performance when using an LSTM-based generator in the 25-block setting is due to the additional network capacity provided by the LSTM. Therefore, in favor of less compute time, our experiments only used an LSTM-based generator for PAIRED in the 25-block setting.

**Student architecture** The student policy architecture resembles the LSTM-based generator architecture, except the student model uses a convolution with 16 filters to embed its partial observation; does not use a random noise vector; and instead of embedding the time step, embeds the student's current direction into a 5-dimensional latent space.

**Choice of hyperparameters** We base our choice of hyperparameters for student agents and generator (i.e. the adversary) on [10]. We also performed a coarse grid search over the student entropy coefficient in $\{0.0, 0.01\}$, generator entropy coefficient in $\{0.0, 0.005, 0.01\}$, and number of PPO epochs in $\{5, 20\}$ for both students and generator, as well as the choice of including an LSTM in the student and generator policies. We selected the best performing settings based on average return on the validation levels of SixteenRooms, Labyrinth, and Maze over 3 seeds. Our final choices are summarized in 4. The main deviations from the settings in [10] are the choice of removing the generator's LSTM (except for PAIRED with 25 blocks) and using fewer PPO epochs (5 instead of 20). For PLR, we searched over replay rate, $p$, in $\{0.5, 0.95\}$ and level buffer size, $K$, in $\{500, 2000, 4000\}$, temperature $\beta$ in $\{0.1, 0.3\}$, and choice of scoring function in $\{\text{positive value loss}, \text{MaxMC}\}$. The final PLR hyperparameter selection was then also used for PLR$^\perp$ and REPAIRED, except for the scoring function, over which we conducted a separate search for each method.

**Zero-shot levels** We make use of the challenging test mazes in [10]: SixteenRooms, requiring navigation through up to 16 rooms to find a goal; Labyrinth, requiring traversal of a spiral labyrinth; and Maze, requiring the agent to find a goal in a binary-tree maze, which requires the agent to successfully backtrack from dead ends. To more comprehensively test the agent's zero-shot transfer performance on OOD classes of mazes, we introduce Labyrinth2, a rotated version of Labyrinth; Maze2, another variant of a binary-tree maze; PerfectMaze, a procedurally-generated maze environment; and LargeCorridor, another procedurally-generated maze environment, where the goal position is randomly chosen to lie at the end of one of the corridors, thereby testing the agent's ability to perform backtracking. Figure 3 provides screenshots of these mazes.

**Compute** All maze-navigating agents were trained using Tesla V100 GPUs. DR required approximately 40 hours to reach 250 million training steps; minimax, 50 hours; PLR variants, 100 hours; and PAIRED variants, 160 hours. In total, our main experimental results, across 25-block and 50-block runs, required roughly 18,300 hours (around 763 days) of training.

## D.2  CarRacing

**Environment details** Each track consists of a closed loop around which the student agent must drive a full lap. In order to increase the expressiveness of the original CarRacing, we reparameterized the tracks using Bézier curves. In our experiments, each track consists of a Bézier curve [22] based on 12 randomly sampled control points within a fixed radius, $B/2$, of the center of the $B \times B$ playfield. The

track consists of a sequence of $L$ polygons. When driving over each previously unvisited polygon, the agent receives a reward equal to $1000/L$. The student additionally receives a reward of -0.1 at each time step. Aligning with the methodology of [18], we do not penalize the agent for driving out of the playfield boundaries, terminate episodes if the agent drives too far off track, and repeat every selected action for 8 steps. The student observation space consists of a $96 \times 96 \times 3$ pixel observation with RGB channels with a clipped, egocentric, bird's-eye view of the vehicle centered horizontally in the top $84 \times 96$ portion of the frame. The remaining $12 \times 96$ portion of the frame consists of the dashboard visualizing the agent's latest action and return. Note that despite the lossiness of the downsampled dashboard, our hyperparameter sweep for the best PPO settings found that including the full frame enabled better performance. Given this observation, the student then decides on a 3-dimensional continuous action, where the components correspond to control values for steer (torque, in $[-1.0, 1.0]$), gas (acceleration, in $[0.0, 1.0]$), and brake (deceleration, in $[0.0, 1.0]$).

**Level generation** Starting from an empty track, the adversary generates a sequence of 12 control points, one per time step, spaced within a fixed radius, $B/2$ of the center $O$ of the playfield. The agent always begins centered at the track polygon closest to $0°$ relative to $O$, facing counterclockwise.

**Generator architecture** At each time step, the generator policy receives the set of all control points so far generated, the current time step encoded as a one-hot vector, and a 16-dimensional random noise vector. The control points are spatially encoded in a $10 \times 10$ grid, called the *sketch*, representing a downsampled and discretized version of the playfield bounds within which the generated track resides. Choosing a control point then corresponds to selecting one of the cells in this grid. After the control points are chosen, each control point's cell coordinates are upscaled to match the original playfield scale. This ensures no two control points are too close together, preventing areas of excessive track overlapping. The sketch is embedded using two $2 \times 2$ convolutions using a stride length of 1 with 8 and 16 channels respectively, each followed by a ReLU layer. The flattened outputs of this sequence of convolutions is then concatenated with an 8-dimensional embedding of the time step and the random noise vector. This combined embedding is then fed through two fully connected layers, each with a hidden size of 256, where the first is followed by a ReLU activation, to produce the policy logits over the 100 choices of control points. Note that we mask out any cells in the sketch that have already been chosen to prevent double selection of the same control point. We also experimented with outputing continuous, downsampled control points in $[0.0, 1.0]$ by learning the $\alpha$ and $\beta$ parameters of a Beta distribution for each of $x$ and $y$ coordinates instead of categorical logits, but found this latter parameterization led to slower learning of generator policies, where the generator policy tended to remain close to or revert to an approximately uniformly random policy.

**Student architecture** The student policy architecture is based on the competitive PPO implementation in [18], which was used as a baseline for AttentionAgent in [38]. This architecture consists of an image embedding module composed of a stack of 2D convolutions with square kernels of sizes 2, 2, 2, 2, 3, 3, channel outputs of 8, 16, 32, 64, 128, 256, and stride lengths of 2, 2, 2, 2, 1, 1 respectively, resulting in a 256-dimensional image embedding. The image embedding is then passed through a fully connected layer with a hidden size of 100, followed by a ReLU layer. This latter output is then fed through two separate fully-connected layers, each with hidden size of 100 and output dimension equal to the action dimension, followed by softplus activations. We then add 1 to each component of these two output vectors, which serve as the $\alpha$ and $\beta$ parameters respectively for the Beta distributions used to sample each action dimension. When training the student, we normalize rewards by dividing rewards by the running standard deviation of returns so far encountered.

**Choice of hyperparameters** To determine the best hyperparameters for the student agents, we performed a grid search, in which we trained a student agent with domain randomization for 300 PPO updates. The grid search covered PPO learning rate in $\{0.001, 0.0003\}$, $\lambda_{\text{GAE}}$ in $\{0.0, 0.5, 0.9\}$, number of PPO epochs in $\{4, 8\}$, PPO number of minibatches per epoch in $\{2, 4, 8\}$, value loss coefficient in $\{0.5, 2.0\}$, whether to grayscale frames, whether to crop frames (i.e remove the dashboard portion), and whether to normalize returns. Further, we found entropy regularization tended to hurt performance of the student policy. Similar to the sharing of PPO hyperparameters between student and generator in [10], we then shared the best PPO hyperparameters for the student with the generator, with the exception of searching over separate choices for the entropy coefficient in $\{0.0, 0.01\}$. We selected the best performing settings based on average return on the validation levels of F1-Italy, F1-Singapore, and F1-Germany over 3 seeds. For PLR, we searched over replay rate, $p$, in $\{0.5, 0.95\}$, level buffer size $K$, in $\{500, 2000, 4000, 8000\}$, replay prioritization in $\{\text{rank}, \text{proportional}\}$, staleness coefficient $\rho$ in $\{0.3, 0.7\}$, and replay distribution temperature $\beta$ in

{0.1, 1.0, 2.0}. The best settings for PLR were then shared with REPAIRED and REPAIRED, except for the scoring function, over which we performed a separate search for each method.

**Zero-shot levels** Our zero-shot levels are based on 20 real-world Formula One (F1) tracks designed to challenge professional racecar drivers. We predominantly selected tracks based on recent F1 seasons, including some historical favorites such as the Nürburgring Grand Prix.[2] This collection of tracks, which we call CarRacing-F1, provides a new benchmark for testing robustness and zero-shot generalization in a continuous control setting. Importantly, these tracks are strictly out-of-distribution and of higher complexity with respect to the training levels, as they cannot be represented by Bézier curves limited to 12 control points. Moreover, each F1 track requires more time steps to solve (1500 or 2000) than allotted for the training tracks (1000). Table 5 provides per-track descriptions, and Figure 19 shows bird's-eye views of each track.

**Compute** All car racing agents were trained on Tesla V100 GPUs. DR and PLR variants required approximately 18 hours to reach 5 million training steps, while PAIRED variants, 24 hours. Our experiments in this domain entailed a total of roughly 9,600 hours (around 400 days) of training.

Table 5: Descriptions for each track in the CarRacing-F1 benchmark.

| Environment | Real-world track | Max episode steps |
|---|---|---|
| Australia | Albert Park | 1500 |
| Austria | Red Bull Ring | 1500 |
| Bahrain | Bahrain International Circuit | 2500 |
| Belgium | Circuit de Spa-Francorchamps | 1500 |
| Brazil | Autódromo José Carlos Pace | 2000 |
| China | Shanghai International Circuit | 2500 |
| France | Circuit Paul Ricard | 2000 |
| Germany | Nürburgring | 2000 |
| Hungary | Hungaroring | 2000 |
| Italy | Monza Circuit | 1500 |
| Malaysia | Sepang International Circuit | 2500 |
| Mexico | Autódromo Hermanos Rodríguez | 2000 |
| Monaco | Circuit de Monaco | 1500 |
| Netherlands | Circuit Zandvoort | 2000 |
| Portugal | Algarve International Circuit | 2500 |
| Russia | Sochi Autodrom | 1500 |
| Singapore | Marina Bay Street Circuit | 2000 |
| Spain | Circuit de Barcelona-Catalunya | 2000 |
| UK | Silverstone | 2000 |
| USA | Circuit of the Americas, Austin | 2000 |

---

[2]We chose not to include the Japanese and Canadian Grand Prix due to the overlapping tracks at Suzuka and the Circuit Gilles Villeneuve.

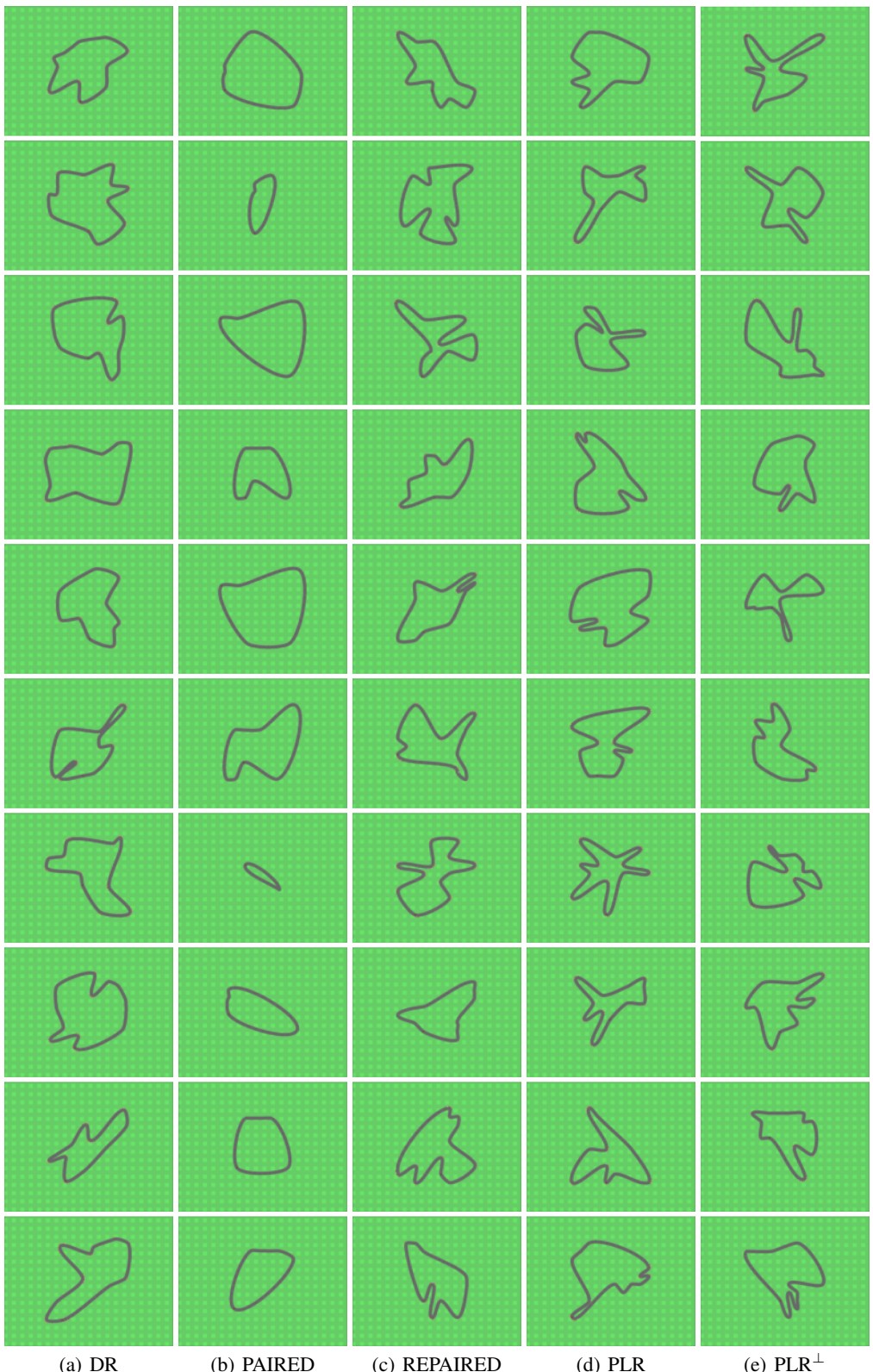

|         |           |              |         |              |
| :-----: | :-------: | :----------: | :-----: | :----------: |
| (a) DR  | (b) PAIRED | (c) REPAIRED | (d) PLR | (e) PLR$^{\perp}$ |

Figure 18: A randomly-selected set of CarRacing tracks generated by each method. (a) Domain Randomization (DR) produces tracks of average complexity, with few sharp turns. (b) PAIRED often overexploits the difference in the students, leading to simple tracks that incidentally favor the antagonist. (c) REPAIRED mitigates this degeneracy, recovering track complexity. (d) PLR and (e) PLR$^{\perp}$ similarly generate tracks of considerable complexity, by prioritizing the most challenging randomly generated tracks.

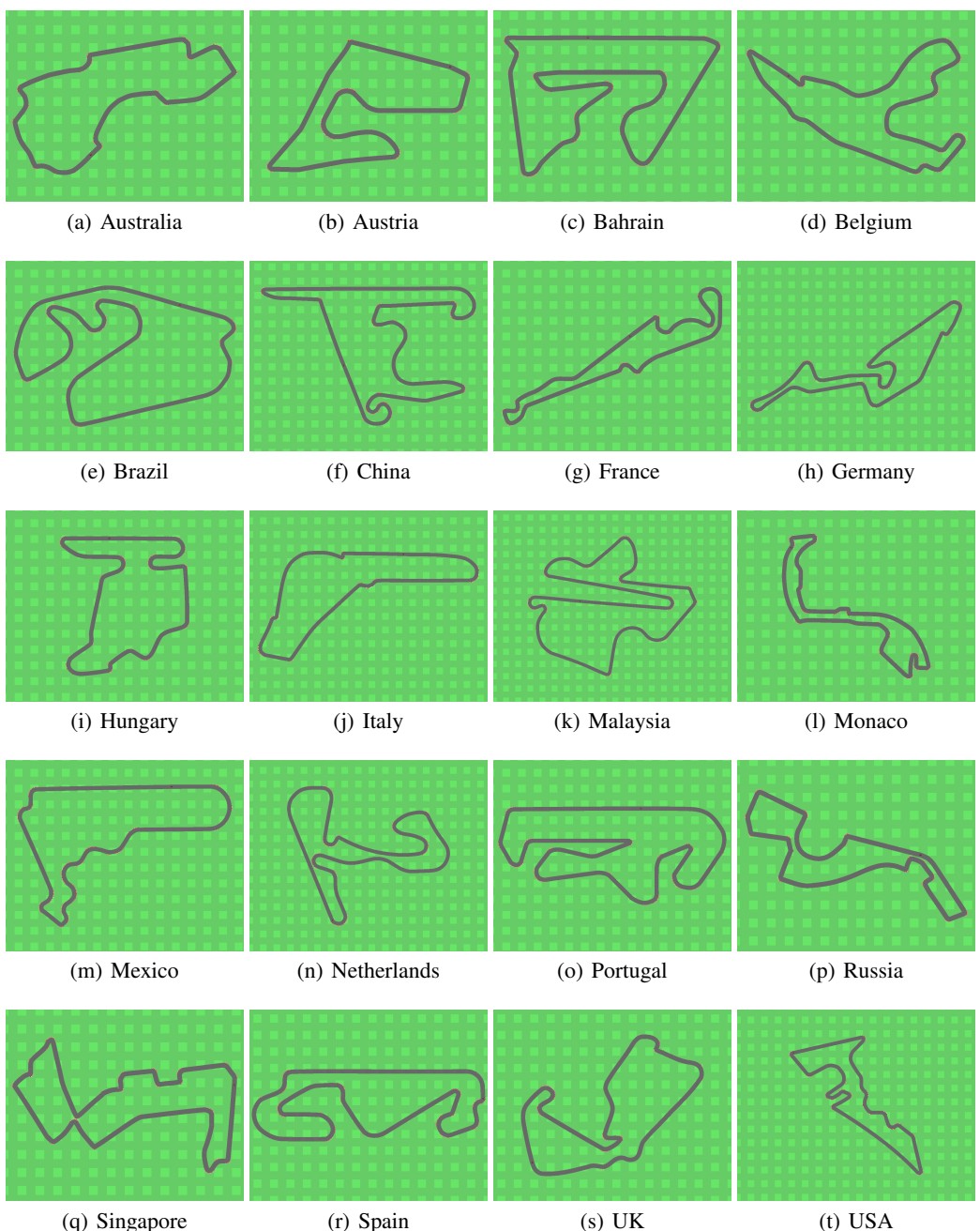

Figure 19: All tracks in the CarRacing-F1 benchmark used for evaluating zero-shot generalization.