# OpenReview forum: "Replay-Guided Adversarial Environment Design"
_NeurIPS.cc/2021/Conference — NeurIPS 2021 Poster_

### Official Review · Reviewer_amNM · 2021-07-14

**Rating:** 7
**Confidence:** 4

**Summary:**

In this paper, the authors propose a common framework,  Dual Curriculum Design, that augments PAIRED with a PLR-based replay mechanism, named as REPAIRED. The theory also suggests convergence to Nash Equilibrium should be assisted by training on fewer data when using PLR—namely by only taking policy-gradient updates from data that originates from the PLR buffer, and only using samples from the environment distribution to populate the buffer. The authors further conducted experiments on a maze environment and on car-racing tracks. Results showed that the proposed approaches outperformed the PLR and other baselines in general.

**Ethical Concerns:**

No ethical concerns.

**Limitations And Societal Impact:**

No negative social concerns.

**Main Review:**

Strengths:
The paper provides theoretical analysis on the robustness guarantees for PLR method and REPAIRED.
The performance of the new proposed algorithms in both experiments is better than the state-of-the-art benchmarks.

Concerns:
The analysis is based on approximate Nash equilibria. The robustness is based on the accuracy of this approximation. However, the theoretical guarantee shows $\pi$ is $4Bp$ close to having optimal worst-case regret. How significant is this distance in the landscape of the policy? It seems to me, both $B$ and $p$ could be big so I have concerns of the effectiveness of the analysis. It will be nice to analyze the sensitivity of this distance to the effectiveness for robustness analytically and experimentally.

Minor things: Some notations are confusing:
1. Line 107, in the left of the equation, the regret is on $\pi_A$ and $\pi_B$, on the right of the equation, the values become $\pi_P$ and $\pi_A$. Actually, throughout the whole paper, the mix of using $\pi_P$ and $\pi_B$ might confuse the readers. It will be great if the notation could be consistent.
2. It will be nice if the authors could give further qualitative and numerical insights to the various performance with different maps. Particularly, why DR is the best among all the methods. Could this unveil some limitations of the proposed method/all PLA-based methods?

**Time Spent Reviewing:**

10 hours

---

> ### Author Response · Authors · 2021-08-08
> **Thank you for your positive review and useful feedback**
>
> We greatly appreciate the time the reviewer has invested in reviewing our paper. Thank you for your positive comments on our work. We are glad you find our theoretical and empirical results substantial.
>
> In response to the concerns and minor comments you raise:
>
> - As you mention, the $4Bp$-Nash equilibrium can mean that the distance to the desired equilibrium can be large as $B$ and $p$ grow, where here $p$ corresponds to the probability of playing the random teacher, which is not optimizing for the student's regret. However, note that we directly operationalize this bound to formulate PLR$^{\perp}$, which deliberately takes no gradient steps on levels sampled from the random generator (i.e. uniform distribution over levels corresponding to domain randomization), and instead only performs gradient steps on levels sampled from the PLR buffer. In doing so, we effectively set $p$ to 0, meaning PLR$^{\perp}$ recovers the original Nash equilibrium corresponding to that in which the student policy achieves a minimax regret policy. Thus, as you can see, our purpose for presenting this bound was to show that it can be used to motivate this specific robustifying adjustment to PLR, producing PLR$^{\perp}$, which provably has a minimax regret policy guarantee at Nash equilibrium. This is an important detail for us to communicate clearly, so we appreciate that you asked for clarification on this. We will make sure this detail is more clearly emphasized in our final manuscript.
> - Thank you for spotting this typo on Line 107 and more generally the usage of $A, B$ subscripts in addition to $P, A$ subscripts in the paper. We have made this notation consistent in our updated manuscript.
> - Regarding the comment on why DR is the best among the methods, we believe the reviewer might have partly misinterpreted the presented results: In both maze and car racing domains, our methods outperform DR. This is shown in Figure 4, Figure 6, and Table 3 in the Appendix. To more clearly emphasize this point, we will add DR test curves to Figure 4 (in place of the horizontal lines marking the performance of the original DR baseline from Dennis et al, 2020 at 3B steps of training), in addition to adding zero-shot evaluation results for DR to the maze results in Figure 3 and Table 2.
> - Additionally, we have further benchmarked each method in a more challenging setting in which each method can only place up to 25 blocks (instead of 50) during training, while evaluating generalization on a wider array of out-of-distribution test mazes. In this more challenging setting, our methods still outperform the baselines across the expanded set of out-of-distribution test levels (summarized in the table below). We have also updated our manuscript with these additional results.
>
> | Method (25 block budget) | Mean solved rate on all OOD mazes |
> | ------------------ | ---------------- |
> | PAIRED         | 0.27&pm;0.03 |
> | REPAIRED (Ours) | 0.41&pm;0.04 |
> | PLR                 | 0.58&pm;0.05 |
> | PLR$^\perp$ (Ours)            | **0.73&pm;0.05** |
>
> We hope that our responses have addressed your concerns, and that you will continue to support our paper for acceptance at NeurIPS.

---

### Official Review · Reviewer_Xs8e · 2021-07-16

**Rating:** 7
**Confidence:** 3

**Summary:**

This paper considers two forms of curriculum design, unsupervised environment design and prioritized experience replay, and highlight their complementary nature.
It then considers the games induced by both these approaches and combines them into a dual curriculum game where the agent faces one teacher or the other with certain probability. Further, the paper analyzes the equilibria of this joint game and the equilibria of the two component games.

The paper then proposes two new algorithms, a robust version of PLR that does not learn on experienced trajectories but only on trajectories sampled by the PLR teacher, and a replay-enhanced PAIRED that improves the UED technique with PLR.

The various approaches are then compared on mazes and car racing domains.

**Limitations And Societal Impact:**

Societal impact is not discussed. The checklist asserts that this discussion is not applicable to their research. However some discussion of curriculum design and its impact on learning agents might have been useful.

**Main Review:**

## Technical Positives:
* This paper show how UED/PAIRED and PLR can be seen as dual approaches and also how they can be combined.
* These approaches fall into the larger space of automatic curriculum design for agents, and explore how different approaches can be combined effectively.
* The game theoretic analysis of the equilibria of the combined game compared with the equilibria of the base games is shown to be an effective approach to analyze such a combination.
* Notably, analyzing PLR through this lens allows principled modifications to the prioritization scheme and the agent update scheme.
* REPAIRED seems to actually be the more explicit combination of the two approaches considered, and the successful combination of PLR and PAIRED is encouraging.
* Despite the other errors alluded to in the supplementary material, overall the experiments on the maze and car racing domains show a fair bit of success for the above combinations. Both PAIRED and PLR seem to benefit from the modifications suggested.

## Questions, Remarks and Drawbacks:
* It seems the supplementary material and the main text are out of sync. On page 7 the main text repeatedly refers to a Figure 7 and the related text does not seem to match up with the discussion. Similarly in Section 6.2 Figure 11 is referred to as a visualization of the test tracks, which should probably refer to Figure 12.
* The assertion that PLR and UED are dual problems seemed unclear and imprecise.
* This "duality" also seems to not have much bearing on the game that is formulated for further analysis.


**Time Spent Reviewing:**

6

---

> ### Author Response · Authors · 2021-08-08
> **Thank you for your positive feedback and additional comments**
>
> Many thanks to the reviewer for their positive review of our paper. We are glad you found our game-theoretic analysis and newly proposed methods to be effective.
>
> In response to your questions and additional remarks:
>
> - Thank you for pointing out this typo. The "Figure 7" referenced in the discussion should be "Figure 4," which shows the zero-shot solve rate of SixteenRooms, Labyrinth, and Maze throughout the course of training. Also thank you for pointing out that "Figure 11" referred to in Section 6.2 should instead refer to "Figure 12," which shows all 20 Formula One tracks we used for zero-shot evaluation.
>
> - We want to clarify the meaning of "dual" used in our work. In our paper, "dual" simply refers to the fact that there are two parallel curricula—one controlled by the curator and the other, by the generator. It is not the intention to use "dual" in the sense of "dual problem" as used in linear programming or "geometric duality" as used in projective geometry. We will make sure our *purely descriptive* usage of "dual" to mean simply two curricula is made clear in our final manuscript.
>
> We believe our paper is improved by your comments, and hope you will continue to support our paper for acceptance at NeurIPS.

---

### Official Review · Reviewer_eYYy · 2021-07-17

**Rating:** 6
**Confidence:** 2

**Summary:**

This paper considers a recent proposed problem, unsupervised environment design, that provides an environment selection scheme to enable the policy being trained under a curriculum learning setting. The proposed method combines two new approaches in this direction, PAIRED and PLR, where the former one learns to generate an environment while the latter one does not include a generator but instead gives a sampling scheme to choose previously encountered environments. The combined method is explained as a two-teacher dual curriculum game and explained theoretically. Experiments demonstrate the effectiveness of the combined method compared to PAIRED and PLR, respectively.

**Limitations And Societal Impact:**

Yes

**Main Review:**

Some comments:

1. The proposed method combines two new approaches, PAIRED and PLR, in the scope of environment designed and replay sampling, where the former one learns to generate an environment while the latter one does not include a generator but instead gives a sampling scheme to choose previously encountered environments. The combined method is explained as a two-teacher dual curriculum game and explained theoretically. I think such an explanation is reasonable and also the experiments are well-designed and sufficiently demonstrate the effectiveness of the combined method to me.

2. The algorithm in Algorithm 1 implies that the combined method alternatively chooses PAIRED and PLR, relying on the sampled d. How is P_D(d) obtained? Moreover, PAIRED is chosen only when d=0. What is the distribution of P_D(d)?

3. In section 6.1, the teacher works by placing one obstructing block per time step and finally places the agent and the goal. This is similar to the implementation of PAIRED, but in which the agent and the goal are placed first. Are there any differences of the designed orders for the teacher? Moreover, it is not very clear to me that as the placement is sequentially decided by the teacher, is this sequential decision progress treated as an MDP process and solved by RL? If so, it is a bit confused for me that in this sequential decision progress, there is no interaction between the teacher and the agents, and therefore the entire designment of the maze can be completed offline, such that the sequential decision problem can be transferred to an one-step decision (generating the entire maze). Of course, generating the entire maze in one step is impractical to be encoded in an action space in RL, and then the problem is cast to train a maze (environment) generator. I am not sure if my understanding is correct, so would the authors provide more explanations on this?


**Time Spent Reviewing:**

4

---

> ### Author Response · Authors · 2021-08-08
> **Thank you for your positive feedback and requests for clarification**
>
> We thank the reviewer for their positive comments on our paper, notably that the reviewer found our explanations clear and that our experiments were well-designed and sufficiently demonstrated the effectiveness of our methods.
>
> Further, thank you for raising some good questions (points 2 and 3 in your review), which we now aim to address in turn:
>
> 2. The distribution $P_D$ is a Bernoulli parameterized by a hyperparameter $p$, as done in the original PLR paper (Jiang et al, 2021). More simply, it means that at the start of each rollout batch, with probability $p$, we sample from the generator (either DR or a learned generator as in the case of REPAIRED), and with probability $1-p$, we sample from the replay distribution controlled by PLR. In our experiments, we perform a hyperparameter sweep over $p$ and report results for each method using the best setting of $p$ found on our set of validation environments. We will aim to ensure that it is perfectly clear that $P_D$ is parameterized as a Bernoulli in the final manuscript.
>
> 3. We used exactly the implementation of the teacher rollouts as employed by PAIRED (our implementation directly makes use of the open-sourced code in the PAIRED repository). Through correspondence with the authors of PAIRED, they produced the strongest results with PAIRED using the parameterization that placed the goal and agent last, after block placement.
>
> You are correct that the environment generation process (e.g. placing blocks, goal, and agent in the MiniGrid maze setting) is modeled as an MDP. In particular, the details of this MDP for the maze and car racing environments are detailed in Appendices D.1 and D.2 respectively. The teacher is then trained as an RL policy on this MDP such that the generated levels maximize the regret of the main student agent. This is the same formulation as used in PAIRED (Dennis et al, 2020). You can view this as parameterizing the level generator as an RL policy, thereby generating levels in what can be a potentially large, complex design space autoregressively. Further, you are correct that we could alternatively cast the problem as training some other kind of generative model to produce levels, but such an optimization would necessarily still require performing RL since the regret objective would not be directly differentiable with respect to the generative model's parameters. We stuck with the RL formulation used in PAIRED as it allows us to compare to this important baseline. That said, we wholeheartedly agree that these are exciting directions for future research.
>
> We hope that in light of these clarifications, you would consider raising your score in support of our paper. If not, we would appreciate if you could clearly state what stands between us and a higher score, in your opinion, after having addressed the two points of information above.

---

> ### Author Response · Authors · 2021-08-19
> **Checking in with Reviewer eYYy**
>
> Dear Reviewer eYYy, we would be grateful if you could confirm whether our clarifications on the parameterization of $P_D$​​​ and the implementation details of the teacher rollouts have addressed your concerns.

---

### Official Review · Reviewer_HQZ9 · 2021-07-17

**Rating:** 6
**Confidence:** 4

**Summary:**

This paper proposes to theoretical analysis of the Prioritized Level Replay (PLR) from prior work and proposes an environment generation framework named Dual Curriculum Design (DCD). The proposed framework is composed of teacher agents that keep generating new environments and a student agent that learns to solve the generated environments. The teacher agents are co-adapted to generate new environments by maximizing the student's agent's regret. In a grid world domain and the CarRacing domain, the proposed method outperforms previous baselines such as PAIRD.

**Limitations And Societal Impact:**

It would be better if the authors could provide more discussions and explanations about why the proposed method cannot be evaluated in continuous domains in this paper.

**Main Review:**

Strengths:

-  Based on prior work, this paper proposes novel theoretical analysis as well as algorithmic solutions for generating environments as curricula.

- The proposed technical solution and the theoretical analysis seem to be reasonable to me.

-  The paper is clearly written and easy to follow.

- This paper could provide interesting insights to future work on environment generation. However, the significance of the proposed method is not entirely convincing due to missing several important baselines.

Weaknesses:

- Several state-of-the-art curriculum learning methods should be compared as baselines, including  [Portelas et al. CoRL 2019], [Florensa et al. ICML 2018], and [Zhang et al. NeurIPS 2020]. It is also important to include the baseline that uniformly samples environments as curricula.

- In Figure 4 and Figure 5, it seems that the learning curves often suffer from noticeable performance regressions during the later stage of training. Could the authors provide more detailed explanations about this phenomenon?

- In some subfigures in Figure 4 (e.g. "SixteenRooms" and "Maze"), the learning curves are cut off before convergence is reached.

- The GridWorld and the CarRacing domains are a little toyish. It would be better to include more challenging task domains for continuous space control as in [Portelas et al. CoRL 2019] and [Florensa et al. ICML 2018].

**Time Spent Reviewing:**

3 hours

---

> ### Author Response · Authors · 2021-08-08
> **Responses to your encouraging feedback and suggestions**
>
> We thank the reviewer for their feedback, and for noting the clarity of our work and how it provides interesting insights for future work in environment generation. Regarding the reviewer's additional points:
>
> - Re additional baselines: We believe our primary contributions are independent of the additional baselines requested by the reviewer: We defined a new theoretical framework (DCD) that predicts algorithmic improvements to PLR and REPAIRED, whose benefits have been empirically validated in our experiments. Therefore, the proper baseline for PLR$^{\perp}$ is PLR, and the proper baseline for REPAIRED is PAIRED.
>
> - Two of the additional baselines mentioned by the reviewer are designed for the goal-conditioned setting, which is not the focus of our work. In fact, directly adapting these methods to our setting is non-trivial and would constitute new research in its own right: Zhang et al, 2020 requires training ensembles of explicitly goal-conditioned value functions, which makes a direct translation impossible. Similarly, Florensa et al, 2018 assumes a goal-conditioned policy in addition to requiring explicit domain knowledge of the UPOMDP's minimum and maximum returns, which is then used to label goals for training their GAN. Additionally, full level generation with GANs is known to be a challenging, open problem (some initial, promising ideas are explored in Schrum et al, 2021's paper on Hybrid Encoding). Further, Florensa et al, 2018 assumes a continuous goal space for which goals vary in terms of reward function, while in our setting, we do not condition on goals, and each UPOMDP instance effectively varies not just the reward function but also the state space and transition dynamics.
>
> - We have adapted the remaining baseline proposed by the reviewer, Portelas et al, 2019 (ALP-GMM), to our CarRacing environment. Unlike our method, ALP-GMM requires domain knowledge about the minimum and maximum return achievable across all levels, for the purpose of normalizing the return when computing the learning progress metric, which is then used in constructing the space over which the GMM is fit. It also assumes a continuous level parameterization. We find that ALP-GMM results in suboptimal policies in our car racing domain, likely because it is incentivized to sample tasks increasing the absolute change in return, which may be satisfied by sampling tasks that reduce the return, leading to similar learning dynamics to PAIRED (which overexploits the antagonist) in this domain. We show a comparison of ALP-GMM against our methods at 5M steps of training below:
>
>   | Method             | Mean F1 return  |
>   | ------------------ | --------------- |
>   | ALP-GMM							| -70 &pm; 11|
>   | DR                 | 322 &pm; 18     |
>   | PAIRED             | 87 &pm; 15      |
>   | REPAIRED (Ours)    | 320 &pm; 16     |
>   | PLR                | 292 &pm; 12     |
>   | PLR$^\perp$ (Ours) | **405 &pm; 11** |
>
> - It is further worth noting that our methods focus on a different problem than Zhang et al, 2020 and Florensa et al, 2018: Unlike these two methods, which aim to improve in-distribution sample efficiency, our work focuses on developing methods with provably robust properties in generalizing to highly out-of-distribution levels.
>
> - We respectfully disagree that our environment is too simple compared to those featured in Portelas et al, 2019 and Florensa et al, 2018. We summarize the differences in environment complexity based on observation dimensionality and number of free parameters specifying each level (or task). Note that both our environments are higher dimensionality than those considered in these papers. In particular, our car racing setting requires learning from pixels, unlike the environments in these other studies. We believe it was valuable to benchmark our methods on both maze and car racing, as it allowed us to test our methods in both discrete and continuous control settings.
>
>   | Env  | Observation dimensionality |Number of level (or task) parameters|
>   | ---- | ---- |---|
>   |  BipedalWalker (Portelas et al, 2019)  | 24 |6|
>   | Ant navigation (Florensa et al, 2018) | 41 |2|
>   | Point Mass (Florensa et al, 2018) | $\le$ 5 |$\le$ 5|
>   | **MiniGrid Maze (Ours)** | **147** |**52**|
>   | **Bezier CarRacing (Ours)** | **27,648** |**36**|
>
> - We agree a random curriculum is a useful baseline, which is equivalent to our DR (domain randomization) baseline.
>
> - Re the comment on Figure 4 and 5: We have found there can be mild oscillations in zero-shot test performance on certain out-of-distribution mazes throughout training due to a form of forgetting. As training progresses, the student agents are effectively performing a form of continual learning, and are liable to forget behaviors that were beneficial on certain test mazes. However, over a longer training horizon, we find the overall trend is improvement in zero-shot return on all our evaluation mazes. Below we include a table showing zero-shot performance (solve rate) of student agents trained on each method from 250M &#8594; 500M steps (2x that featured in the curves for Figure 5):
>
>   | Method             | SixteenRooms                          | Labyrinth                                 | Maze                                      |
>   | ------------------ | ------------------------------------- | ----------------------------------------- | ----------------------------------------- |
>   | PAIRED             | 0.60&pm;0.05 &#8594; 0.74&pm;0.05     | 0.29&pm;0.05 &#8594; 0.49&pm;0.06         | 0.15&pm;0.03 &#8594; 0.27&pm;0.04         |
>   | REPAIRED (Ours)    | 0.70&pm;0.05 &#8594; 0.89&pm;0.03     | 0.68&pm;0.05 &#8594; 0.59&pm;0.06         | 0.34&pm;0.05 &#8594; 0.30&pm;0.05         |
>   | PLR                | 0.67&pm;0.05 &#8594; **0.94&pm;0.02** | 0.73&pm;0.06 &#8594; 0.74&pm;0.05         | 0.44&pm;0.06 &#8594; 0.50&pm;0.06         |
>   | PLR$^\perp$ (Ours) | **0.76&pm;0.04** &#8594; 0.82&pm;0.04 | **0.91&pm;0.03** &#8594; **0.89&pm;0.04** | **0.51&pm;0.05** &#8594; **0.58&pm;0.06** |
>
> - Finally, regarding the comment on why certain subfigures in Figure 4 are truncated before convergence: Our intention was to focus on those subfigures (showing solved rate vs. # PPO updates) in the early stages of training to emphasize how quickly our methods improve zero-shot test sample efficiency, with a relatively small number of PPO update cycles. We will gladly include the same plot extended over the full training horizon of 500M steps for the 50-block setting in our updated manuscript. Note that these are the same plots as the primary figure in Figure 4, but with the x-axis set to the number of student gradient updates (as PLR$^\perp$ and REPAIRED perform fewer updates than the other methods).
>
> We hope you find our responses reasonably address all of your concerns and that you will consider raising your support for our work to an "Accept."

---

> > ### Comment · Reviewer_HQZ9 · 2021-08-27
> > **Response to the Authors**
> >
> > I appreciate the authors' efforts in including the additional experiments and providing further explanations. Therefore, I've raised my score to 6.
> >
> > Regarding Figure 4, I would suggest the authors provide figures that show the full curves across a longer time horizon in the appendix.

---

> > > ### Author Response · Authors · 2021-08-30
> > > **Thank you for your response**
> > >
> > > We are glad that our additional experiments and clarifications addressed the reviewer's concerns, and thank the reviewer for their score increase. We will provide the extended curves with a longer time horizon in the camera-ready.

---

> ### Author Response · Authors · 2021-08-19
> **Checking in with Reviewer HQZ9**
>
> Dear Reviewer HQZ9, we would be grateful if you could confirm whether our responses have sufficiently addressed your concerns. To summarize our responses,
>
> - We emphasized that the primary contribution of this paper is independent of the additionally proposed baselines by the reviewer, and the proper baselines for PLR$^\perp$ and REPAIRED are PLR and PAIRED respectively.
> - We underscored why the methods introduced in Florensa et al, 2018 and Zhang et al, 2020 are not directly applicable to the setting studied in our work.
> - In accordance with the reviewer’s suggestion, we reported the performance of ALP-GMM (Portelas et al, 2019) on our car racing environment. Our results show that it underperforms the other methods in this domain.
> - We analyzed the alternative environments proposed by the reviewer and showed that they are not clearly more complex than those studied in our work. In fact, in terms of observation and environment dimensionality, ours are more complex.
> - To address the concern around the longer-term trend of the zero-shot test return curves, we reported improved zero shot performance for our methods after 500M steps of training—twice that reported in our original submission.

---

### Official Review · Reviewer_58ME · 2021-07-30

**Rating:** 5
**Confidence:** 3

**Summary:**

This paper studies the problem of automatically designing a distribution of environments that adjusts to the learning agent, called unsupervised environment design. Specifically, they propose Dual Curriculum Design in which two co-evolving teachers, i.e. one is a generator and another one is a replay teacher, introduce new levels/environments to the learning agent (a.k.a student) to learn a policy that can be effectively generalized to new and unseen environments. Besides, this paper proposes a new theory about PLR and REPAIRED robustness guarantees. The paper evaluates zero-shot generalization, emergent complexity, and scalability of their proposed method in multiple environments and scenarios.

**Limitations And Societal Impact:**

This paper discusses an approach for unsupervised environment design based on existing methods for toy/simulated environments ​which doesn't have a social impact. As such, I don't think this work has any negative societal impact.

**Main Review:**

While this paper studies an important and interesting problem of automatically designing the distribution of environments for a learning agent, unfortunately, the proposed method(s) is confusing and the main contribution of the paper is not clear and I don't fully understand what are the takeaways from this paper. Namely:

       1. The main idea of the paper is built on top of PAIRED where a replay teacher is added to it and called REPAIRED. Although REPAIRED results are not good and not consistent across all experiments, this could have been the main story of the paper.

       2. PLR and PLR+ ( for simplicity, I use PLR+ instead of PLR^\perp): The discussions, theory, changes to PLR, and finally introduction of PLR+ are quite out of place in this paper. If the PLR+ is the contribution of this paper, then the paper should be about this not REPAIRED. It seems the authors decided to include both since neither one shows good results in all experiments and each doesn't have a significant contribution alone.

Writing, results, and presentation: It is quite hard to follow the paper and get a clear picture of the experiments:

       3. What does robustness mean here? Are you referring to generalization? or something else?

       4. Why are some methods used in some experiments but not in others? For instance, PLR+ is not in Figure 3, or part of Figure 4?

       5. While I applauded authors to call out their bug in Figure 6, I am not sure what can I conclude about this paper since sometimes REPAIRED shows better results and some other time PLR+., take Figure 6 as an example, REPAIRED is better in F1-Italy and PLR+ is good in F1-Singapore and F1-Germany (both methods shows similar performance for CarRacing-v0 though)

       6. What is B in line 164?

       7.  What is the relationship between \pi* and \pi in Eq1? What is \pi*?

       8. What does context mean in 145? "However, to formalize this game, we must first formalize the single-teacher context. Suppose the UPOMDP is clear from context."

       9. What is \theta_A in line 106? Is it a learnable parameter? Is it \pi_A parameter? or something else?

       10. Why maximum-regret objective (U^R) and uniform objective (U^U) are defined in line 152?   They are not referred to or used in section 3 and only referred later in section 4.1

       11. Where is Figure 7 in the main text? (lines 264 and 283) Figure 7 in the appendix talks about something else and doesn't tell anything about replay rate of 0.5 for example.

       12. What are the free parameters of an underspecified environment? what underspecified means (line 4)?

       13. When this method generates a level, what are the mechanisms in place to make sure that there exists a solution and that level can be solved?


In summary, the significance and technical contributions of this work are not clear, the paper needs significant revision, and it is not ready for this conference. I'd highly recommend focusing on one concrete story in the paper rather than try to run two different narratives.



**Time Spent Reviewing:**

4.5

---

> ### Author Response · Authors · 2021-08-08
> **Responses to your useful and detailed feedback**
>
> We thank the reviewer for their comments, and for noting that our work addresses an important and interesting problem. We believe all of the reviewer’s concerns around clarity of the paper can be addressed easily as they are minor issues that can be fixed in a sentence or two (as we demonstrate below). We thank the reviewer for making us aware of these points, which will help us improve the paper so that such misunderstandings are avoided for other readers in the future. Below, we address the reviewer’s points in turn:
>
> Re 1 and 2: We explicitly state in the abstract and in the introduction that our work "reveals a natural class of UED methods we call Dual Curriculum Design (DCD). Crucially, DCD includes both PLR and PAIRED as special cases. This connection allows us to develop novel theory for PLR, providing a version with a robustness guarantee...in addition to demonstrating that PLR improves the performance of PAIRED, from which it inherited its theoretical framework." We will revise the abstract and introduction to put an even stronger focus on how our framework of DCD connects PLR and PAIRED under a single theoretical framework, allowing principled improvements to both algorithms (having PLR train on less data leading to PLR$^\perp$ and combining PLR-based replay with PAIRED leading to REPAIRED).
>
> Re 3: We use robustness and zero-shot generalization in the same way as prior work, notably Dennis et al, 2020 (NeurIPS 2020), which introduced the PAIRED algorithm. Robustness refers to the minimax regret property of our method—that at equilibrium, our methods provably achieve a minimax regret policy that optimally trades off regret across all levels. Generalization is used to mean that the learned policy can solve handcrafted environments that are highly out-of-distribution (OOD) to those in the training distribution.
>
> Re 4: We have added PLR$^\perp$ alongside PLR in both Figure 3 and 4, where PLR$^\perp$ shows a clear gain in zero-shot generalization over PLR. (The tables in response to point 5 below shows how they compare.)
>
> Re 5:  The mean performance of PLR$^\perp$ across the full CarRacing F1 benchmark is statistically significantly greater than that of the other methods, including PLR, at p=0.01. Similarly, REPAIRED is significantly better than PAIRED. Our corrected barplot in Figure 6 reports this result, which we summarize in the table below.
>
> | Method             | Mean F1 return  |
> | ------------------ | --------------- |
> | DR                 | 322 &pm; 18     |
> | PAIRED             | 87 &pm; 15      |
> | REPAIRED (Ours)    | 320 &pm; 16     |
> | PLR                | 292 &pm; 12     |
> | PLR$^\perp$ (Ours) | **405 &pm; 11** |
>
> Likewise, Figure 4 shows that PLR$^\perp$  and REPAIRED significantly improve over PLR and PAIRED respectively. A snapshot of this plot at 250M steps is provided below.
>
> | Method (50 block budget) | SixteenRooms     | Labyrinth        | Maze             |
> | ------------------------ | ---------------- | ---------------- | ---------------- |
> | PAIRED                   | 0.60&pm;0.06     | 0.29&pm;0.05     | 0.15&pm;0.03     |
> | REPAIRED (Ours)          | 0.70&pm;0.05     | 0.68&pm;0.05     | 0.35&pm;0.05     |
> | PLR                      | 0.67&pm;0.05     | 0.72&pm;0.06     | 0.37&pm;0.06     |
> | PLR$^\perp$ (Ours)       | **0.77&pm;0.04** | **0.90&pm;0.03** | **0.51&pm;0.05** |
>
> Note that while the original DR baseline used in the PAIRED paper gave the random level generator a budget of 25 blocks while PAIRED was given 50, we found that if we allow each method to place up to 50 blocks, they all, with the exception of PAIRED, effectively solve the maze domain after 500M steps of training. We thus benchmarked our methods on the more challenging maze setting where our methods can only place up to 25 blocks, testing generalization to the same out-of-distribution levels. On this more challenging setting, after 250M steps of training, PLR$^\perp$ significantly outperforms the other methods in zero-shot transfer on the full set of OOD mazes, after the same number of gradient updates, and REPAIRED similarly outperforms PAIRED (See table below). This additional benchmark shows that our methods are less sensitive to the exact environment parameterization, and can succeed in more challenging generalization settings.
>
> | Method (25 block budget) | Mean return on all OOD mazes |
> | ------------------ | ---------------- |
> | PAIRED         | 0.27&pm;0.03 |
> | REPAIRED (Ours) | 0.41&pm;0.04 |
> | PLR                 | 0.58&pm;0.05 |
> | PLR$^\perp$ (Ours)            | **0.73&pm;0.05** |
>
> Re 6: As defined in Line 162, at the start of Theorem 1, $B$ refers to the maximum difference between utilities $U^1_t$ and $U^2_t$.
>
> Re 7: Here, $\pi^*$ refers to the optimal policy for the UPOMDP fully specified with free parameters set to $\theta$ , and $\pi$ refers to the current policy $\pi$ whose utility is evaluated by the function $U^R_t(\pi, \theta)$. We will add this further clarification to the final manuscript.
>
> Re 8: Here "single-teacher context" refers synonymously to "single-teacher setting," i.e. the case of there being only a single teacher agent. We will gladly adjust this phrasing in our final manuscript to "single-teacher setting". In the sentence "Suppose the UPDOMP is clear from context," context refers to the surrounding discussion.
>
> Re 9: Thank you for pointing out this inconsistent notation. Here, $\theta_A$ denotes the same quantity as $\theta$ in this passage, which is a specific instance of the UPOMDP free parameters chosen by the PAIRED adversary—which defines a specific level of the UPDOMDP. We will correct $\theta_A$ to simply $\theta$, as used in the rest of the passage.
>
> Re 10: The utility $U^R_t$ is used in the definition and proof of Corollaries 1 and 2, and the utility $U^U_t$ is used in the definition and proof of Corollary 1. Corollary 1 is defined in Section 4.1, and Corollary 2, in Section 5. Both proofs are provided in Appendix A. We will make clearer upon defining these utility terms that they will come into play in these subsequent corollaries.
>
> Re 11: Thank you for pointing out this mislabeling. The instances referring to "Figure 7" that you mention should be referring to "Figure 4" instead. We will make sure to correct this in our final manuscript.
>
> Re 12: The free parameters of a UPOMDP are those parameters that vary from level to level, leading to variations across levels—for example the positions of obstacles in the maze environments. This definition and example are provided in Section 2.1, but we can add additional clarity around this to avoid confusion.
>
> Re 13: Note that impossible levels have a maximum regret of 0. Thus, a major benefit of the regret-maximizing teacher featured in our DCD framework is that the regret objective naturally disincentivizes the teacher to generate levels that are not solvable. We will gladly emphasize this point in our final manuscript.
>
> We believe, and hope the reviewer agrees, that we have effectively addressed all of the reviewer's concerns. In particular, we hope our response clarifies that the contribution of our paper is the introduction of the DCD framework and the theory underlying DCD (Theorem 1) that led to principled improvements to two previous UED algorithms, as supported by Corollaries 1 and 2. Furthermore, we believe the other excellent points raised by the reviewer can be easily fixed in the final version of the manuscript as follows:
> - Points 1,2,6,8,12 seem to be due to simple misunderstandings that we will preemptively address in the paper.
> - Points 3,7,9,10,11, and 13 are easily addressed through clarifying sentences in our final manuscript similar to our responses above.
> - Points 4 and 5 are clearly addressed by our existing PLR$^{\perp}$ results, as well as our additional results on the more challenging setting of MiniGrid mazes with 25 blocks, which we will show in Figures 3 and 4.
>
> We hope that the reviewer will consider increasing their support for our paper given these clarifications and the required minor modifications that will improve our paper.

---

> > ### Comment · Reviewer_58ME · 2021-08-24
> > **Thanks for your responses.**
> >
> > Thanks for your comments. Your responses indeed clarified the paper further for me.
> > While I think my initial score was a bit harsh ( hence I increased it), I stand by my comments about PLR+ and REPAIRED. I just can't wrap my head around the contribution of the paper and not sure what the contribution of the paper is. If you have just written the paper about PLR+, we would have different discussions here, but I don't see the connection between PLR and PAIRED leads to developing a new theory for PLR and why REPAIRED is even discussed as the main contribution in the paper. This is the main issue for me and has not been addressed in your responses and your response basically was repeating the same thing as the paper abstract.

---

> > > ### Author Response · Authors · 2021-08-24
> > > **Regarding your remaining concerns**
> > >
> > > Thank you for raising your score. We are glad that our clarifications were useful in addressing many of your concerns. It seems your outstanding concern is that you are not clear about why REPAIRED is presented alongside PLR$^\perp$. We will further elucidate this below, and also re-emphasize what we stated above: it is simply not true that PLR$^\perp$ and REPAIRED don’t work consistently, or don’t return good results (please refer to our answer to points 4 and 5 in our initial response). We would be grateful if you could explain to us, modulo the clarification presented below, and in light of our other individual answers to your points, why you feel the paper is worth rejecting if both of our theoretically-motivated methods outperform the previous state-of-the-art baselines in this class of UED algorithms.
> > >
> > > We would like to reemphasize that PLR$^\perp$ is the main contribution of our paper, and it is a direct consequence of uniting the theory behind PLR and PAIRED. This argument lies at the center of our work and our entire paper is structured so to reflect this argument: We start by establishing Theorem 1 (Section 3), which we show leads to Corollary 1, which motivates PLR$^\perp$ (Section 4) and Corollary 2, which motivates REPAIRED (Section 5). We don’t claim in our paper that REPAIRED is our main contribution, as emphasized by our structuring of Section 5 as only a brief anecdotal section following the primary discussion in Section 3 and 4. The algorithm REPAIRED simply falls directly out of Corollary 2. As it is the direct combination of PLR and PAIRED, we include it as a DCD baseline that uses both a generator and curator to compare its effectiveness with PLR$^\perp$, which, in contrast, is a DCD algorithm with only the curator. However, the main contribution of this paper is PLR$^\perp$, as highlighted in our abstract, introduction, and discussions, as well as in our previous response.
> > >
> > > We are happy to, on the basis of your feedback, modify the abstract and intro in order to further clarify this. If you have suggestions as to where in these sections (or elsewhere) the writing makes things unclear relative to the explanation we provide above, we would love to hear them and incorporate them into the paper. We have already incorporated improvements based on your previous feedback (which we addressed in our previous response) in our updated paper. Thank you for helping us improve the clarity of the paper thus far.

---

> ### Author Response · Authors · 2021-08-19
> **Checking in with Reviewer 58ME**
>
> Dear Reviewer 58ME, we would be grateful if you could confirm whether our responses have addressed your concerns. To summarize our responses,
>
> - We provided simple clarifications that address each point of confusion in 1, 2, 6, 8, and 12.
> - We provided straightforward answers to each of the questions raised in points 3,7,9,10,11, and 13.
> - In response to points 4 and 5, we highlighted that PLR$^\perp$​ does in fact consistently and statistically significantly outperform the other methods on both domains in our study. In emphasis of this fact, we included the table recapping the car racing results in Figure 5, as well as an additional table showing that PLR$^\perp$​​ significantly outperforms the other methods on the maze domain. We have now included the corresponding curve for PLR$^\perp$ alongside the other methods in the plot shown in Figure 4.
> - We have updated our paper with these straightforward clarifications and minor corrections in response to your feedback.

---

### Decision · Program_Chairs · 2021-09-27

**Decision:**

Accept (Poster)

**Comment:**

The paper proposes a unified approach for environmental design in RL.

The reviewers positively evaluated the paper and recommended for acceptance.

AC.